

**Impact of ambient conditions on the Si isotope fractionation in marine pore**
**fluids during early diagenesis**
Sonja Geilert[1], Patricia Grasse[1], Kristin Doering[1,2], Klaus Wallmann[1], Claudia Ehlert[3], Florian Scholz[1],
Martin Frank[1], Mark Schmidt[1], Christian Hensen[1]
[1]GEOMAR Helmholtz Centre for Ocean Research Kiel, Wischhofstr. 1-3, 24148 Kiel, Germany
[2]Department of Oceanography, Dalhousie University, Halifax, Canada
[3]Marine Isotope Geochemistry, ICBM, University of Oldenburg, Germany
ABSTRACT
Benthic fluxes of dissolved silica (Si) from sediments into the water column are driven by the
dissolution of biogenic silica ($bSiO_2$) and terrigenous Si minerals and modulated by the precipitation
of authigenic Si phases. Each of these processes has a specific effect on the isotopic composition of
silica dissolved in sediment pore waters such that the determination of pore water $\delta^{30}Si$ values can
help to decipher the complex Si cycle in surface sediments. In this study, the $\delta^{30}Si$ signatures of pore
fluids and $bSiO_2$ in the Guaymas Basin (Gulf of California) were analyzed, which is characterized by
high $bSiO_2$ accumulation and hydrothermal activity. The $\delta^{30}Si$ signatures were investigated in the
deep basin, in the vicinity of a hydrothermal vent field, and at an anoxic site located within the
pronounced oxygen minimum zone (OMZ). The pore fluid $\delta^{30}Si_{pf}$ signatures differ significantly
depending on the ambient conditions. Within the basin, $\delta^{30}Si_{pf}$ is essentially uniform averaging
+1.2±0.1‰ (1SD). Pore fluid $\delta^{30}Si_{pf}$ values from within the OMZ are significantly lower (0.0±0.5‰,
1SD), while pore fluids close to the hydrothermal vent field are higher (+2.0±0.2‰, 1SD).
Reactive transport modelling results show that the $\delta^{30}Si_{pf}$ is mainly controlled by silica dissolution
($bSiO_2$ and terrigenous phases) and Si precipitation (authigenic aluminosilicates). Precipitation
processes cause a shift to high pore fluid $\delta^{30}Si_{pf}$ signatures, most pronounced at the hydrothermal
site. Within the OMZ however, additional dissolution of isotopically depleted Si minerals (e.g. clays)
facilitated by high mass accumulation rates of terrigenous material ($MAR_{terr}$) is required to promote
the low $\delta^{30}Si_{pf}$ signatures while precipitation of authigenic aluminosilicates seems to be hampered by
high water/rock ratios. Guaymas OMZ $\delta^{30}Si_{pf}$ values are markedly different from those of the
Peruvian OMZ, the only other marine setting where Si isotopes have been investigated to constrain
early diagenetic processes. These differences highlight the fact that $\delta^{30}Si_{pf}$ signals in OMZs worldwide
are not alike and each setting can result in a range of $\delta^{30}Si_{pf}$ values as a function of the environmental




conditions. We conclude that the benthic silica cycle is more complex than previously thought and that additional Si isotope studies are needed to decipher the controls on Si turnover in marine sediment and the role of sediments in the marine silica cycle.

KEYWORDS: Si isotopes, reverse weathering, hydrothermal system, oxygen minimum zone, environmental conditions

1.  INTRODUCTION

Silicon (Si) is one of the key macronutrients in the ocean mainly utilized by siliceous organisms such as diatoms, radiolarians or sponges (see recent review by Sutton et al., 2018). The marine Si cycle is closely linked to the carbon (C) cycle by marine siliceous organisms, which transport C to the sediment and thus exert a strong control on C export from the atmosphere impacting present and past climate (e.g. Lewin, 1961; Tréguer and Pondaven, 2000; Tréguer and De La Rocha, 2013; and recent reviews by Frings et al., 2016 and Sutton et al., 2018). Studies of Si isotopes ($\delta^{30}$Si) have revealed complex uptake and dissolution processes of siliceous organisms, which have a dominant control on  the $\delta^{30}$Si distribution in ocean waters (e.g. de la Rocha et al., 1997; Varela, 2004; Cardinal et al., 2005; Beucher et al., 2008; Fripiat et al., 2011; Ehlert et al., 2012; Grasse et al., 2013; Sutton et al., 2013; de Souza et al., 2014, 2015). Diatoms constitute the largest part of the Si cycling fluxes in the ocean (Ragueneau et al., 2000) and discriminate between its isotopes during Si uptake, whereby the light isotopes are preferentially incorporated into the diatom frustules (e.g. de la Rocha et al., 1997). Si isotope fractionation during silica uptake is dependent on e.g. the diatom species, the availability of Fe, and the degree of silica utilization. Fractionation factors between -0.5 and -2.1‰ have been derived from regional water mass mixing and laboratory studies (de la Rocha et al., 1997; Varela, 2004; Cardinal et al., 2005; Beucher et al., 2008; Sutton et al., 2013; Meyerink et al., 2017).

After a planktonic bloom whereupon the nutrients are exhausted, biogenic silica ($bSiO_2$, mainly diatoms) sinks through the ocean, partially dissolves and accumulates on the seafloor, where its preservation and recycling is controlled by dissolution and Si re-precipitation processes. The dissolution of $bSiO_2$ mainly controls the accumulation of silicic acid ($Si(OH)_4$) in pore fluids, although the in-situ concentration remains below the equilibrium concentration of dissolved $bSiO_2$, which has been explained by simultaneous formation of authigenic silicates (e.g. McManus et al., 1995; Van Cappellen and Qiu, 1997a, b; Rickert et al., 2002). This process is termed reverse weathering given that the authigenic precipitates are rich in seawater-derived cations, like Na, K or Mg (Mackenzie et al., 1981; Michalopoulos and Aller, 1995). Experimental studies of $bSiO_2$ dissolution kinetics revealed a dependence of the $bSiO_2$ reactivity on sediment depth as well as the ratio between terrigenous



material and $bSiO_2$ (Michalopoulos and Aller, 1995; Van Cappellen and Qiu, 1997a, b; Dixit et al.,
2001; Rickert et al., 2002). Marine weathering of terrigenous material (primary silicates like feldspars
and secondary silicates like clays), was found to release cations such as aluminum (Al) or iron (Fe),
which reduce the solubility and dissolution rate of $bSiO_2$ and induce aluminosilicate precipitation
(Michalopoulos and Aller, 1995; Van Cappellen and Qiu, 1997a, b; Michalopoulos et al., 2000; Dixit et
al., 2001; Rickert et al., 2002; Loucaides et al., 2010). The accumulation of Si in pore fluids and the Si
reflux into bottom waters are controlled by three interdependent processes, namely: opal
dissolution, dissolution of terrigenous solids, precipitation of authigenic minerals.
Early silica diagenesis has been shown to fractionate Si isotopes as a function of the crystallization
state, seawater Si concentration, sedimentation rate, and terrigenous mineral content (Tatzel et al.,
2015; Geilert et al., 2016). Since the light $^{28}Si$ isotope is more reactive compared to the heavier
isotopes ($^{29}Si$, $^{30}Si$) , processes such as  reverse weathering, adsorption, and direct silica precipitation
from saturated solutions show low $\delta^{30}Si$ values in the reaction product and high $\delta^{30}Si$ values in its
substrate (i. e. fluids) (e.g. Georg et al., 2006a, 2009; Delstanche et al., 2009; Opfergelt et al., 2013;
Geilert et al., 2014; Roerdink et al., 2015; Ehlert et al., 2016). The few modelling and experimental
studies, addressing Si isotope fractionation during formation of secondary phases, report isotope
fractionation factors between -1.6 and -2‰ (Ziegler et al., 2005; Méheut et al., 2007; Dupuis et al.,
2015; Ehlert et al., 2016).
The $\delta^{30}Si$ data for marine pore waters from the Peruvian margin upwelling region, which is
characterized by very high diatom productivity and $bSiO_2$-rich sediments (Abrantes et al., 2007;
Bruland et al., 2005), agree with these findings and clearly indicate Si isotope fractionation as a
consequence of authigenic aluminosilicate precipitation accompanied by a Si isotope fractionation
factor of -2.0‰ (Ehlert et al., 2016). Also pore fluids from the Greenland margin and Labrador Sea
reflect early diagenetic reactions detected by pore fluid $\delta^{30}Si$ values (between +0.76 and +2.08‰)
and the $\delta^{30}Si$ values were interpreted as the product of reverse weathering reactions (Ng et al.,
2020). In this study, the Guaymas Basin in the Gulf of California was chosen as study area, as it is
characterized by a relatively high diatom productivity and sediments that are predominantly
composed of diatomaceous muds (up to 50% diatoms; Kastner and Siever, 1983). Moreover, the
Guaymas Basin in the Gulf of California is influenced by hydrothermal activity (e.g. Von Damm, 1990).
These $bSiO_2$-rich sediments are thus ideal for dedicated studies of early diagenesis under the
influence of different thermal and redox conditions. We investigated the processes controlling Si
isotope fractionation during early diagenesis based on pore fluid and $bSiO_2$ data from three
fundamentally different environmental settings within the Guaymas Basin including the deep basin, a
hydrothermal site, and a site within the Oxygen Minimum Zone (OMZ) on the slope of the Guaymas
Basin (Fig. 1). In addition, the Si isotope composition of the water column, bottom waters, and



hydrothermal fluids was determined. A numerical transport-reaction model was applied to the OMZ
setting to constrain marine weathering processes and to compare the results to the Peruvian margin.
The aim of this study was to constrain the factors controlling Si isotope fractionation during early
diagenesis and to identify processes influencing $bSiO_2$ dissolution and authigenic silicate
precipitation.

2.  GEOLOGICAL SETTING, SAMPLING, AND METHODS

2.1 Geological setting


The Guaymas Basin in the Gulf of California is a currently opening continental rifting environment
with two graben systems (northern and southern trough), which are offset by a transform fault and
reaches spreading rates of up to 6 cm yr$^{-1}$ (Calvert, 1966). High biological productivity and
terrigenous matter input result in high sediment accumulation rates and have produced thick
sequences of organic-rich sediments (DeMaster, 1981; Curray and Moore, 1982). Siliceous sediments
at the Guaymas Slope show fine laminations within the OMZ (dissolved oxygen <10 µM at ca. 500 –
900 m water depth; Campbell and Gieskes, 1984) due to the absence of burrowing organisms
(Calvert, 1964). The Guaymas Basin is characterized by vigorous hydrothermal activity represented
by Black Smoker type vents discovered in both the northern and southern troughs (Berndt et al.,
2016; Von Damm et al., 1985). Hydrothermal plumes spread horizontally and mix with deep basin
water up to 300 m above the seafloor resulting in a fraction of hydrothermal fluids of ~0.1% in the
deep waters of the Guaymas Basin (Campbell and Gieskes, 1984). Hydrothermal sills and dikes
intruding into the sediments were found to accelerate early diagenetic reactions (due to the released
heat) and change pore fluid geochemistry significantly (Gieskes et al., 1982; Kastner and Siever, 1983;
Von Damm et al., 1985; Von Damm, 1990). At present, pore fluids in surface sediments show a
seawater composition (Geilert et al., 2018) and the absence of diagenetic high-temperature
processes render these pore fluids suitable for studying recent early diagenetic processes.

2.2 Sampling


Sediments were sampled via multicorer (MUC) deployment during RV SONNE cruise SO241 in
summer 2015 as described in detail in Geilert et al. (2018). In total, 6 stations have been investigated;
4 within the basin (termed basin sites: MUC33-11, MUC22-04, MUC23-05, MUC15-02), one in the
vicinity of a hydrothermal vent field (termed hydrothermal site: MUC66-16), and one within the OMZ
(termed OMZ site: MUC29-09) (Fig. 1, Table 1). The coring locations within the basin were sampled in



water depths between 1726 and 1855 m below sea level (mbsl). The hydrothermal site (MUC66-16)
was sampled in 1842 mbsl and was located at a distance of ~500 m from the active hydrothermal
mound described by Berndt et al. (2016). The OMZ site (MUC29-09) was sampled on the Guaymas
Basin slope of the Mexican mainland in 665 mbsl.
After core retrieval, bottom water above the sediment was sampled and filtered immediately using
0.2 µm cellulose acetate membrane filters. Bottom water from MUC22-04 may have been
contaminated with surface waters during core retrieval as indicated by Si and Mn concentrations
(53.8 µM and 0.05 µM, respectively) lower than at the remaining sites within the deep basin, which
show distinct anomalies caused by mixing with hydrothermal plume fluids (e.g. MUC15-02: Si = 177.8
µM and Mn = 0.34 µM). Processing of sediments was conducted in a cool laboratory in an argon-
flushed glove bag immediately after core retrieval. Sampling intervals were 1-5 cm with the highest
resolution close to the sediment surface and increasing distance downcore. Pore fluids were
separated from sediments by centrifugation (20 min at 4500 rpm) and subsequently filtered (0.2µm
cellulose acetate membrane filters) for further analyses.

Water column samples were taken using a video-guided Niskin Water sampler CTD-Rosette System.
Water samples were taken in the basin above MUC15-02 at 1844 mbsl (VCTD02), within the
hydrothermal plume between 1781 and 1800 mbsl (VCTD06 and 09) above the Black Smoker mound
as described in Berndt et al. (2016), and above the OMZ site (MUC29-09) at 586 mbsl (Fig. 1).

2.3 bSiO$_2$ separation and digestion

The bSiO$_2$ mass fractions of sediment samples were determined using an automated leaching
method following Müller and Schneider (1993) at GEOMAR Helmholtz-Centre for Ocean Research
Kiel. The sample material was treated with 1M NaOH at 85°C to extract the opal fraction. The
increase in dissolved silica was monitored and evaluated using a method described by DeMaster
(1981). The precision of the mass fraction determination was 5 to 10% (1SD). The bSiO$_2$ was
separated from the sediment for Si isotope analyses following the method of Morley et al. (2004).
About 500 mg of a freeze-dried sediment sample was transferred into a centrifuge tube. Organic
matter and carbonate material was removed by adding H$_2$O$_2$ (Suprapur) and HCl, respectively. Clay
particles (grain size <2µm) were separated from the remaining sediment by the Atterberg method
following Stoke's law (Müller, 1967). The remaining sediment (bSiO$_2$ and heavy minerals) was sieved
using a 5 µm-sieve and subsequently separated from the remaining detritus using heavy liquid
separation (sodium-polytungstate solution). The heavy liquid purification method was repeated until
clean, examined via light microscopy, bSiO$_2$ samples (>95%) were obtained. Light microscopy



revealed that the bSiO$_2$ fraction essentially consisted of diatoms and only traces of radiolarians and
sponges were present (<5%). The bSiO$_2$ samples were stored in Milli-Q water (MQ water). The bSiO$_2$
sample of the OMZ site stems from a nearby gravity core (GC07), which is described in detail in
Geilert et al. (2018). Aliquots of the cleaned bSiO$_2$ samples were transferred into Teflon vials and
dried on a hot plate. Subsequently, 1 ml of 0.1 M NaOH was added and the samples were placed on a
hot plate at 130°C for 24 hours. After sample digestion, the supernatant and residue (undissolved
traces of radiolarians and sponges) were separated via centrifugation. The supernatant was treated
with 200 µl H$_2$O$_2$ (Suprapur) in order to remove remaining organic matter and then dried and re-
dissolved in 1 ml 0.1 M NaOH at 130°C for 24 hours. After the digestion procedure, the samples were
diluted with MQ water and neutralized with 1 M HCl.

2.4 XRD measurements


X-ray diffraction analyses of the dried clay samples were performed at the Central Laboratory for
Crystallography and Applied Material Science, ZEKAM, dept. of Geosciences, University Bremen,
using a Philips X'Pert Pro multipurpose diffractometer. The diffractometer was equipped with a Cu-
tube, a fixed divergent slit of $\frac{1}{4}$°, a secondary Ni filter, and a X'Celerator detector system. A
continuous scan from 3 to 85° 2θ was applied for the measurements with a calculated step size of
0.016° 2θ (calculated time per step was 50 seconds). Quantification of mineral phases were based on
the Philips software X'Pert High Score$^{TM}$, the freely available X-ray diffraction software MacDiff 4.25
(Petschick et al., 1996), and the QUAX full-pattern method after Vogt et al. (2002). The standard
deviation is ±1-3% for well crystallized minerals (see also Vogt et al., 2002) and ±5% for the remaining
mineral phases.

2.5 Geochemical analyses of fluid and solid phases


Analyses of major and trace element concentrations of pore fluids from the basin sites and the
hydrothermal site as well as the water column are described in Geilert et al. (2018). OMZ site pore
fluids were treated in the same way. In brief, the pore fluids were analysed onboard by photometry
(NH$_4$) and on shore for dissolved anions (Cl) and cations (Si, K, Na, Mg) using ion chromatography (IC,
METROHM 761 Compact, conductivity mode) and inductively coupled plasma optical emission
spectrometry (ICP-OES, VARIAN 720-ES), respectively. Analytical precision was constrained using the
IAPSO seawater standard for all chemical analyses (Gieskes et al., 1991) and was found to be <1% for
Cl, <2% for K, Na, Mg, and <5% for Si.



Freeze dried and ground sediment samples were digested in HF (40% Suprapur), $HNO_3$ (Suprapur),
and $HClO_4$ (60% p.a.) for major element analyses. The accuracy of the method was tested by method
blanks and the reference standards SDO-1 (Devonian Ohio Shale, USGS) and MESS-3 (Marine
Sediment Reference Material, Canadian Research Council). The digested samples were measured for
their K and Al contents by ICP-OES (VARIAN 720-ES) and reproducibility was ≤5%. Total carbon (TC)
and total organic carbon (TOC) were measured in freeze-dried and ground sediment samples by flash
combustion using the Carlo Erba Element Analyzer (NA-1500). Carbonate carbon ($CaCO_3$) was
calculated by subtracting TOC from TC.

The digested $bSiO_2$ samples were analyzed for their Al and Si contents using the Agilent 7500 series
quadrupole ICPMS at GEOMAR to provide information about potential clay contamination of the
separated $bSiO_2$ fraction (Shemesh et al., 1988), whereby Al/Si ratios below 50 mM/M are considered
as negligible clay contamination (van Bennekom et al., 1988; Hurd, 1973). Al/Si ratios in the studied
$bSiO_2$ ranged between 15 and 39 mM/M, with three exceptions in MUC22-04, MUC15-02, and
MUC66-16 yielding Al/Si ratios of 71 mM/M, 57 mM/M, 50 mM/M, respectively. However, all
$\delta^{30}Si_{bSiO2}$ values agreed well with surrounding $\delta^{30}Si_{bSiO2}$ values and clay contamination is thus
considered insignificant. All other $bSiO_2$ samples are considered clay-free.

2.6 Sample purification and Si isotope measurements

Fluid and digested $bSiO_2$ samples were prepared for Si isotope measurements following the
purification method of Georg et al. (2006b). The concentration of the samples was adjusted and
loaded (1 ml with ~64 µM Si) onto 1 ml pre-cleaned cation-exchange resin (Biorad AG50 W-X8) and
subsequently eluted with 2 ml MQ water. Matrix effects originating from dissolved organic
compounds and anions, which cannot be separated by this purification method, have previously
been found to potentially influence Si isotope measurements (van den Boorn et al., 2009; Hughes et
al., 2011). However, no influence of the matrix effects on pore fluid Si isotope measurements has
been found during our measurements following several tests described in Ehlert et al. (2016). Briefly,
Ehlert et al. (2016) removed organic compounds by $H_2O_2$ and and $SO_4$ by Ba addition yielding $\delta^{30}Si$
values identical to untreated samples, within error. Therefore, our samples were not treated with
$H_2O_2$ or Ba before sample purification and Si isotope measurements.
Si isotope samples were measured in medium resolution on a NuPlasma MC-ICPMS (Nu
InstrumentsTM, Wrexham, UK) at GEOMAR using the Cetac Aridus II desolvator. Sample Si
concentrations of about 21 µM resulted in a $^{28}Si$ intensity of 3 to 4 V. The MQ blank was ≤ 3mV,
resulting in a blank to signal ratio <0.1 %. The measurements were performed using the standard-



sample bracketing method to account for mass bias drifts of the instrument (Albarède et al., 2004). Si
isotopes are reported in the $\delta^{30}$Si notation, representing the deviation of the sample $^{30}$Si/$^{28}$Si from
that of the international Si standard NBS28 in permil (‰). Long-term $\delta^{30}$Si values of the reference
materials Big Batch (-10.6±0.2‰; 2SD; n=49), IRMM018 (-1.5±0.2‰; 2SD; n=48), Diatomite
(+1.3±0.2‰; 2SD; n=44), and BHVO-2 (-0.3±0.2‰; 2SD; n=13) are in good agreement with $\delta^{30}$Si
values in the literature (e.g. Reynolds et al., 2007; Zambardi and Poitrasson, 2011). The seawater
inter-calibration standard Aloha (1000 m) resulted in +1.3±0.2‰ (2SD; n=8) in very good agreement
to Grasse et al. (2017). Additionally, two in-house matrix standards have been measured. The pore
fluid matrix standard yielded an average $\delta^{30}$Si value of +1.3±0.2‰ (2SD; n=17) and the diatom matrix
standard (E. rex) -1.0±0.2‰ (2SD; n=22), which agrees well with earlier reported values (Ehlert et al.,
2016). All samples were measured 2-4 times on different days and the resulting $\delta^{30}$Si values have
uncertainties between 0.1 and 0.4‰ (2SD, Table 1). The $\delta^{30}$Si values of pore fluids, bSiO$_2$, and bottom
water are given as $\delta^{30}$Si$_{pf}$, $\delta^{30}$Si$_{bSiO2}$, and $\delta^{30}$Si$_{bw}$, respectively. Error bars in the figures indicate the
uncertainty of the individual sample measurements (two standard deviation, 2SD).

2.7 Numerical model


A numerical reactive-transport model was applied to simulate Si turnover within OMZ site sediments.
The model was based on a  previously published Si isotope model (Ehlert et al., 2016) and was
extended to consider the dissolution of additional phases. A detailed description of the model can be
found in the supplementary information.

2.8 Calculation of the amount of terrigenous material and mass accumulation rate


The amount of terrigenous material (%) for the Guaymas Basin was calculated as the total mass
subtracted by the carbonate content (CaCO$_3$), the organic matter content (OC), and the bSiO$_2$ content
(Sayles et al., 2001):

Terrigenous material = 100 − (CaCO$_3$ + OC + bSiO$_2$)          (1)


The mass accumulation rate (MAR) was calculated as

$MAR = S * d * (1 - \Phi)$          (2)




with $S$ the sedimentation rate as 0.18 cm yr$^{-1}$ (Thunell et al., 1994) and $d$ the bulk dry density as 2.5 g
cm$^{-3}$. The porosity ($\Phi$) was taken at the deepest part of the core with 0.925.

3.   RESULTS

3.1 Water chemistry and sediment composition

All water column, pore fluid, and hydrothermal Si concentration data, bSiO$_2$ weight fractions as well
as Si isotope values are reported in Tables 1 and 2. A detailed description of the water column
properties, pore fluids, and hydrothermal fluid chemistry can be found in Berndt et al. (2016) and
Geilert et al. (2018). Pore fluids predominantly show a seawater composition at all sampling sites and
are not influenced by high temperature processes related to sill intrusions or mixing with
hydrothermal fluids. Pore fluid geochemistry of major elements in the OMZ resembles that of the
remaining sampling sites with the exception of a strong enrichment in NH$_4$ (Table 1S and Geilert et
al., 2018) as well as high Fe and low Mn concentrations (Scholz et al., 2019). The porosity corrected K
(see supplement) and Al contents in the sediments ranged between 0.9 and 21.2 wt% and 2.9 and
66.6 wt%, respectively (Table 1). The TOC contents ranged between 0.3 and 7.8 % and are shown in
Table 1S.

3.2 Bottom water, water column, and hydrothermal fluid Si concentrations and $\delta^{30}$Si values

The bottom water Si concentration ranged between 173 and 254 μM for all basin sites and the
hydrothermal site (between 1726 and 1855 mbsl) with the exception of MUC22-04, where Si
concentrations were as low as 54 μM (possible surface water contamination, see section 2.2 and
Table 1). The bottom water $\delta^{30}$Si$_{bw}$ signatures ranged between +1.5‰ and +2.0‰ for all basin sites
and the hydrothermal site and overlap within error (average $\delta^{30}$Si$_{bw}$: +1.8±0.2‰, 1SD; highest $\delta^{30}$Si$_{bw}$
for surface contaminated sample (MUC22-04)). Bottom water Si concentration for the OMZ site was
31 μM (665m water depth). The bottom water within the OMZ site had a distinctly lower $\delta^{30}$Si$_{bw}$
value of +0.8‰. Here, a potential contamination with surface waters can be excluded given that they
are characterized by high $\delta^{30}$Si values (from 1.7 to  4.4‰; Ehlert et al., 2012; Grasse et al., 2013), due
to  the preferential biological uptake of $^{28}$Si (de la Rocha et al., 1997).

The basin water (VCTD02), which was sampled about 1 m above the seafloor, had a Si concentration
of 163 μM and a $\delta^{30}$Si$_{deepBasin}$ value of +1.5‰. Hydrothermal plume Si concentrations ranged between
253 and 690 μM and $\delta^{30}$Si values ranged from +0.7‰ (VCTD09-06) to +1.4‰ (VCTD06-06). The water



column within the OMZ (586 mbsl) had a Si concentration of 78 µM and a $\delta^{30}Si_{OMZ}$ value of +1.5‰
(Table 2).

3.3 Pore fluid Si concentration and $\delta^{30}Si_{pf}$ values


Pore fluid Si concentrations generally increased asymptotically with depth from bottom water values
until reaching maximum average concentrations between 605 and 864 µM Si (Table 1, Fig. 2). For the
basin  sites MUC33-11 and MUC15-02 as well as the OMZ site the Si concentrations asymptotically
increased until average values of 742 (≥8 cm below seafloor (cmbsf)), 729 (≥9 cmbsf), and 765 (≥9
cmbsf) µM were reached, respectively. Hydrothermal site Si concentrations were higher and
increased to 864 µM (≥4.5 cmbsf). MUC22-04 and MUC23-05 Si concentrations initially increased to
values of on average 605 µM (5.5 - 11 cmbsf) and 735 µM (3.5 - 9 cmbsf) and then decreased again to
364 µM (26 cmbsf) and 640 µM (22 cmbsf), respectively.

Pore fluid $\delta^{30}Si_{pf}$ values for all basin sites ranged from +0.9‰ to +1.5‰ (Table 1, Fig. 2), which is
lower than the respective bottom water $\delta^{30}Si_{BW}$ values ($\delta^{30}Si_{BW}$ from +1.6‰ to +2.0‰). The decrease
in Si concentration at MUC22-04 and MUC23-05 below 13 cmbsf and 11 cm cmbsf, respectively, is
not reflected in a significant change in $\delta^{30}Si_{pf}$. The hydrothermal site showed the highest $\delta^{30}Si_{pf}$ values
ranging from +1.8‰ and +2.2‰, which is higher than the bottom water $\delta^{30}Si_{BW}$ (+1.5‰). In contrast,
the OMZ site had the lowest $\delta^{30}Si_{pf}$ values between -0.5‰ and +0.8‰, which was also characterized
by very low $\delta^{30}Si_{bw}$ (+0.8‰; Table 1).

3.4 bSiO$_2$ content and $\delta^{30}Si_{bSiO2}$ values


The bSiO$_2$ content of the sediments (Table 1, Fig. 2) varied between 4.7 wt% and 47.6 wt%. Lowest
contents were present at the basin site MUC22-04 (7.6 -13.1 wt%) and the hydrothermal site (4.7 -
14.6 wt%). The remaining sampling sites showed higher bSiO$_2$ contents of on average 23±7 wt%
(1SD). The $\delta^{30}Si_{bSiO2}$ signatures ranged between +0.4‰ and +1.0‰ and did not vary systematically
with depth or sampling site within error. The small variability in $\delta^{30}Si_{bSiO2}$ signatures most likely stems
from natural variability within the Guaymas Basin.

3.5 XRD results of the clay fraction


The main silicate mineral phases of all samples were phyllosilicates (16-59 wt%), primary silicates
(quartz, plagioclase, potassium feldspar; 15-38 wt%), and amorphous SiO$_2$ (4-43 wt% including



abiogenic and biogenic opal) (Table 2S). The phyllosilicates were mainly composed of variable
fractions of smectite, illite, montmorillonite, and kaolinite. Apart from silicate minerals, minor
amounts of Fe-(hydr-)oxides (≤10 wt%, most between 2 and 3 wt%), pyroxenes (≤8 wt%), and
carbonates (≤6 wt%) were present. Biogenic opal fragments were identified via light microscopy to
be the dominating amorphous silicate phase at all sites besides the hydrothermal site and MUC23-05
in the basin basin. At the hydrothermal site, the abiogenic amorphous silica fraction was the
dominating silica phase in the uppermost and lowermost core sections with only minor occurrences
of biogenic opal fragments. Abiogenic amorphous silica was also found in the uppermost and
lowermost core sections of MUC23-05.

4. DISCUSSION

Pore fluid Si concentration and $\delta^{30}Si_{pf}$ signatures vary significantly between sampling sites and appear
to depend strongly on ambient conditions. The Si concentration and isotope compositions are
proposed to be affected by dissolution of $bSiO_2$, the dissolution of terrigenous phases, and the
formation of authigenic aluminosilicates; the latter process is defined as reverse weathering.
Dissolution of $bSiO_2$ is most effective in the reactive surface layer (≤ 10 cmbsf) where the degree of
silica undersaturation is highest. As soon as Si is released into solution via $bSiO_2$ dissolution, Si re-
precipitates as authigenic aluminosilicates as a function of the availability of reactive metals, made
available by dissolution of terrigenous material (e.g. Michalopoulos and Aller, 1995; Van Cappellen
and Qiu, 1997a,b; Loucaides et al., 2010). In the course of this process authigenic silicate
precipitation induces $\delta^{30}Si$ fractionation, whereby the $^{28}Si$ is preferentially incorporated into the solid
phase, enriching the remaining fluid in $^{30}Si$ (e.g. Georg et al., 2006a; Opfergelt et al., 2013; Ehlert et
al., 2016). In the following sections, the processes during early diagenesis influencing pore fluid
$\delta^{30}Si_{pf}$ signatures under different ambient conditions are discussed. For the OMZ site, we quantify
these processes using a reactive transport model and compare the results to the only other OMZ site
where pore fluid $\delta^{30}Si_{pf}$ data is available, the Peruvian margin.

4.1. Influences on $\delta^{30}Si_{pf}$ due to source mixing

In the open ocean, a strong correlation between the inverse Si concentration (1/Si) and Si isotope
composition in intermediate and deep waters exists, showing low $\delta^{30}Si$ values with high Si
concentrations and can be used to identify water mass mixing between two endmembers with
distinct Si characteristics (e.g. de Souza et al., 2012). Here, we use the two endmember mixing Eq. (3)



to calculate the mixing between the deep water column and fluids originating from $bSiO_2$ dissolution
according to

$$\delta^{30}Si_{mix} = \frac{(\delta^{30}Si_{\text{water column}} * [Si]_{\text{water column}} * f) + (\delta^{30}Si_{bSiO2} * [Si]_{bSiO2} * (1-f))}{([Si]_{\text{water column}} * f) + ([Si]_{bSiO2} * (1-f))}$$

(3)

with $\delta^{30}Si_{\text{water column}}$ and $[Si]_{\text{water column}}$ as the respective water column Si isotope composition and
concentration (Table 2) and $\delta^{30}Si_{bSiO2}$ as the average $bSiO_2$ value (+0.8‰) of all sites. The equilibrium
concentration in respect to $bSiO_2$ dissolution was derived from an experimental study by Van
Cappellen and Qiu (1997a) with $[Si]_{bSiO2}$ = 900µM. Mixing fractions are represented by $f$, varied over
100% water column and 0% fluids affected by $bSiO_2$ dissolution and vice versa.
Pore fluid $\delta^{30}Si_{pf}$ values of all sites deviate from mixing curves between the deep water column and
fluids originating from $bSiO_2$ dissolution and are obviously affected by additional processes (Fig. 3).
The $\delta^{30}Si_{pf}$ values of the basin sites and hydrothermal site are higher (+0.9‰ to +2.2‰) compared to
the respective mixing curves, while $\delta^{30}Si_{pf}$ values of the OMZ site are lower (-0.5‰ to +0.8‰). While
a shift to lower $\delta^{30}Si_{pf}$ values compared to fluid mixing point to dissolution of an isotopically light
phase, show higher $\delta^{30}Si_{pf}$ values that precipitation processes are important, given that the light Si
isotope is preferentially incorporated in authigenic secondary phases (Georg et al., 2009). This
indicates that processes governing the pore fluid Si isotope composition differ significantly between
the individual sites.

4.2. Influences on $\delta^{30}Si_{pf}$ from terrigenous and biogenic material


The terrigenous/$bSiO_2$ ratio was found to be the main mechanism controlling asymptotic pore fluid Si
concentration and the benthic Si flux (Van Cappellen and Qiu, 1997a, b). Maximum Si concentrations
were reached asymptotically at four out of six sampling sites (Fig. 2). Two sites within the basin
(MUC22-04 and MUC23-05) show lower Si concentrations in the deep core sections (Fig. 2), which
are most likely related to the decrease of reactive silica with depth, caused by the formation of less
soluble silica phases (Van Cappellen and Qiu, 1997a). At these sites, the asymptotic Si concentration
is defined as the maximum concentration values in the center of the core. The amount of terrigenous
material for the Guaymas Basin was calculated according to Eq. (1) and accounts for 75%.

Asymptotic Si concentrations plotted against the terrigenous/$bSiO_2$ ratio fall on the global trend
except for the hydrothermal site (Fig. 4a). Here, high geothermal gradients are likely responsible for
the higher Si concentrations with respect to the global trend (see also section 4.3.2). In contrast to



the asymptotic Si concentration, no strong correlation of the pore fluid $\delta^{30}Si_{pf}$ values with the
terrigenous/bSiO$_2$ ratio exists (Fig. 4b). In order to identify processes responsible for the different
pore fluid $\delta^{30}Si_{pf}$ values and to facilitate comparison between the three settings within the Guaymas
Basin, only average $\delta^{30}Si_{pf}$ values will be discussed in the following.

Average $\delta^{30}Si_{pf}$ values show distinct variations between the individual settings. The $\delta^{30}Si_{pf}$ values of
the OMZ site are lower (0.0±0.5‰, 1SD, n = 6) than those of the basin sites (+1.2±0.1‰, 1SD, n = 17)
and the hydrothermal site, which shows the highest $\delta^{30}Si_{pf}$ values (+2.0±0.2‰, 1SD, n = 3).

### 4.3 Influence of the ambient environmental conditions on the $\delta^{30}Si_{pf}$ values

#### 4.3.1 Basin sites


Pore fluid $\delta^{30}Si_{pf}$ values of the basin sites deviate from the mixing curve between the deep water
column and fluids originating from bSiO$_2$ dissolution and are shifted to higher values (+1.2±0.1‰; Fig.
3). This shift to higher $\delta^{30}Si_{pf}$ values can be explained by Si re-precipitation as authigenic
aluminosilicates, which preferentially incorporate the light $^{28}Si$ isotope (Fig. 5a).  Alteration of
terrigenous material leads to mobilization and re-precipitation of Al and the uptake of K from
seawater in the authigenic phase (Michalopoulos and Aller, 2004). The sedimentary K/Al ratio can
thus be used to detect these early diagenetic reactions in addition to pore fluid $\delta^{30}Si_{pf}$ values.
Authigenic aluminosilicates formed during alteration of terrigenous material were found to have K/Al
ratios of 0.32 (Michalopoulos and Aller, 2004), which is higher than the pristine K/Al ratio of
terrigenous material carried by rivers (K/Al = 0.19; Viers et al., 2009). The average K/Al ratio of the
basin sites is 0.34±0.01 (1SD), which is in the same range as K/Al ratios  indicative of authigenic
aluminosilicate formation (Michalopoulos and Aller, 2004). Thus, bSiO$_2$ dissolution and re-
precipitation of Si in authigenic aluminosilicates control the pore fluid $\delta^{30}Si_{pf}$ values in the basin sites.

#### 4.3.2 Hydrothermal site


The $\delta^{30}Si_{pf}$ values from the hydrothermal site are higher (+2.0±0.2‰) than the mixing curve between
the deep water column and fluids originating from bSiO$_2$ dissolution and also much higher than pore
fluid $\delta^{30}Si_{pf}$ values from the basin (Fig. 3, 5a). The high $\delta^{30}Si_{pf}$ values indicate that precipitation plays a
significant role at this site.  Sedimentary K/Al ratios are equivalent to basin values (K/Al = 0.34) and
thus show the formation of authigenic aluminosilicates.  Consequently, the higher pore fluid $\delta^{30}Si_{pf}$
values compared to the basin sites can either be explained by a different Si isotope fractionation



factor or by the precipitation of additional silica phases. The hydrothermal site is located in close
proximity to a hydrothermal vent field and hydrothermal deposits are mainly composed of Fe-
sulfides (Berndt et al., 2016), while nearby sediments are dominated by amorphous silica, quartz, and
Fe-Si silicates (e.g. ferrosilite, fayalite) (Kastner, 1982; Von Damm et al., 1985). Si adsorption to iron
(oxyhydr)oxide and incorporation into Fe-Si gels can create substantial Si isotope fractionation
(Delstanche et al., 2009; Zheng et al., 2016). Pore fluids show high $Fe^{2+}$ concentrations (up to 190 µM;
Scholz et al., 2019) and Si concentrations lower than amorphous Si and quartz solubilities at various
depths (this study; Von Damm et al., 1985; Gieskes et al., 1988), indicating precipitation of the
respective mineral phase during pore fluid ascent. Gieskes et al. (1988) reported on amorphous silica
cement in the hydrothermally-influenced sediments of the Guaymas Basin, which is supported by
findings of this study and likely explains the high amorphous $SiO_2$ contents identified by XRD (~35
wt%; see Sect. 3.5; Table 2S). In the Guaymas Basin, high thermal gradients (up to 11 K m$^{-1}$; Geilert et
al., 2018) caused by igneous sill intrusions near the active spreading center significantly influence
diagenetic reactions at depth and accelerate Si dissolution and precipitation (Fig. 5a) (e.g. Kastner
and Siever, 1983). This can also explain the high pore fluid $\delta^{30}Si_{pf}$ values, given that deep Si saturated
fluids ascent and Si precipitates, likely along with Fe, over a large temperature range, whereby lower
temperatures are associated with larger Si isotope fractionation shifting pore fluid $\delta^{30}Si_{pf}$ to the
observed high values (Geilert et al., 2014; Zheng et al., 2016).

4.3.3 OMZ site


At the OMZ site, the $\delta^{30}Si_{pf}$ values are significantly lower (on average 0.0±0.5‰) than the water
column $\delta^{30}Si_{water\ column}$ value (+1.5±0.2‰) and also lower than the $\delta^{30}Si_{bSiO2}$ value (+0.8‰) (Fig. 2, 3).
Interestingly, the only other $\delta^{30}Si_{pf}$ values from an OMZ were obtained at the Peruvian margin (Ehlert
et al., 2016), where the $\delta^{30}Si_{pf}$ values are slightly higher than the water column $\delta^{30}Si_{bw}$ values (+1.6‰
and +1.5‰, respectively). As Ehlert et al. (2016) concluded, the Peruvian $\delta^{30}Si_{pf}$ values are influenced
by $bSiO_2$ dissolution and precipitation of authigenic aluminosilicates, the latter process shifting pore
fluid $\delta^{30}Si_{pf}$ to values higher than those of the water column. Consequently, in order to explain the
extremely low $\delta^{30}Si_{pf}$ values at the Guaymas Basin OMZ, an additional process must occur. We
hypothesize that a phase enriched in $^{28}Si$ needs to dissolve in order to shift the pore fluid $\delta^{30}Si_{pf}$
values and this phase might be 1) iron (oxyhydr)oxides with adsorbed $^{28}Si$ or 2) terrestrial clays.

Silicon exhibits a strong affinity to iron (oxyhydr)oxides (see also section 4.3.2; Davis et al., 2002) and
Si isotopes fractionate significantly during Si adsorption and co-precipitation (Delstanche et al., 2009;
Zheng et al., 2016). Dissolved $Fe^{2+}$ in pore fluids can be transferred across the sediment-water



interface via diffusion and re-precipitates as iron (oxyhydr)oxides, where it subsequently dissolves
again in the reducing sediment. This process can repeat, resulting in multiple cycles of Fe dissolution
and re-precipitation on the Guaymas Basin slope (Scholz et al., 2019). We hypothesize that the light
$^{28}$Si adsorbs to iron (oxyhydr)oxides in the water column and that upon reductive dissolution of Fe
minerals in the surface sediment, the light $^{28}$Si isotope is re-released into the pore fluids, adding to
the observed low $\delta^{30}$Si$_{pf}$ values. However, the quantification of this Fe-Si shuttle and the contribution
to the low $\delta^{30}$Si$_{pf}$ values to the OMZ pore fluids remains difficult given that Fe undergoes multiple
cycles of dissolution and re-precipitation. Furthermore, the exact process of complexation, Si isotope
fractionation, and co-precipitation is unknown and requires further investigations. We can only
speculate that the transport of $^{28}$Si via the Fe-Si shuttle is only of minor importance given that the
MAR of bSiO$_2$ and terrigenous material are dominating the Si supply to Guaymas OMZ sediments
(Calvert, 1966; DeMaster, 1981a).

The low $\delta^{30}$Si$_{pf}$ values can also be explained by dissolution of terrigenous clay particles, which are
generally enriched in $^{28}$Si, showing a large range in $\delta^{30}$Si with the majority between -3‰ to 0‰
(Frings et al., 2016 and references therein). Clays are usually considered to be the stable end product
of silicate weathering. However, fine clay particles and highly reactive surface sites of clays such as
montmorillonite, smectite and illite may dissolve in natural waters (Cappelli et al., 2018; Golubev et
al., 2006; Köhler et al., 2005). The dissolution is promoted by organic ligands and the reduction of
structural iron of clay minerals under reducing conditions (Anderson and Raiswell, 2004). Humic
substances that are abundant in OMZ sediments enriched in organic matter may catalyze the
microbial reduction of structural iron in clays (Lovley et al., 1998) and their dissolution (Liu et al.,
2017). Clays are abundant in OMZ sediments, given that fine-grained terrigenous material is
transported downslope from the shelf to the basin (Scholz et al., 2019). Furthermore, the microbial
oxidation of ferrous Fe in these fine-grained silicate minerals and its subsequent conversion to
reactive iron minerals was also found to contribute to the Fe cycling at the Guaymas Basin slope
(Scholz et al., 2019).

In order to constrain the possibility of terrigenous clay dissolution and the related shift to low $\delta^{30}$Si$_{pf}$
values in the Guaymas OMZ a reactive transport model was applied, based on our previously
published $\delta^{30}$Si model (Ehlert et al., 2016). The data obtained at the Guaymas OMZ were used to
model the turnover of Si in these sediments and the previously published model extended to
consider additional processes (Fig. 6). A full description of the model is presented in the
supplementary information. The model was fit to dissolved silica concentrations and $\delta^{30}$Si values
measured in pore fluids and biogenic opal and K/Al ratios determined in the solid phase (Fig. 6). High



rates of terrigenous clay dissolution were applied at the sediment surface to reproduce the observed
minima in $\delta^{30}Si_{pf}$ pore fluid values and K/Al ratios in a model run best fitting our data set (Fig. 6, best
fit). Since the terrigenous phases deposited at the sediment surface contain potassium ($K_{terr}$ = 1.7
wt%, K/Al = 0.19; Viers et al., 2009) and are depleted in $^{30}Si$ (clay $\delta^{30}Si$ ≈ -2‰; Frings et al., 2016 and
references therein), the dissolution of these phases induces a decline in pore fluid $\delta^{30}Si$ and solid
phase K/Al (supplementary information). The precipitation of authigenic phases that are depleted in
$^{30}Si$ (Si isotoppe fractionation: $\Delta^{30}Si_{au}$ = -2‰; Ehlert et al., 2016) and characterized by high K contents
(K/Al = 0.32; Michalopoulos and Aller, 2004) induces a down-core increase in pore fluid $\delta^{30}Si_{pf}$ and
solid phase K/Al below the surface minimum. Consequently, terrigenous clay dissolution under the
reducing conditions of the OMZ and subsequent authigenic aluminosilicate precipitation can explain
the low $\delta^{30}Si_{pf}$ values detected in Guaymas OMZ pore fluids (Fig. 5).

Additional simulations were conducted to investigate how the solid phase and pore fluid composition
is affected by the dissolution of terrigenous clay phases and the precipitation of authigenic phases
(Fig. 6). The surface minima in pore fluid $\delta^{30}Si_{pf}$ and solid phase K/Al disappear when the dissolution
rate is set to zero ($R_{terr}$ = 0) while the ongoing precipitation of authigenic phases leads to a strong
down-core increase and high values at depth that are not consistent with the data. Pore fluid $\delta^{30}Si_{pf}$
and solid phase K/Al values strongly decrease with depth when the rate of authigenic phase
precipitation is set to zero ($R_{au}$ = 0) such that the model yields values that are significantly lower than
the measured values. Dissolved silica concentrations cannot be used to further constrain $R_{terr}$ and $R_{au}$
because they are largely controlled by the dissolution of biogenic opal ($R_{opal}$). Dissolved K
concentrations show a much lower sensitivity to $R_{terr}$ and $R_{au}$ than solid phase K/Al ratios due to the
high porosity of the OMZ sediments. Changes in dissolved K are largely eliminated by molecular
diffusion that is favored by the high porosity while the effect of the solid phase reactions $R_{terr}$ and $R_{au}$
on the pore fluid composition is diminished by the low solid phase contents and the high background
concentration of dissolved K in ambient bottom waters.  However, the model runs show that the
more sensitive pore fluid $\delta^{30}Si_{pf}$ and solid phase K/Al can be used to constrain the balance between
the dissolution of terrigenous phases and the precipitation of authigenic phases and that both
reactions are required to model the low $\delta^{30}Si_{pf}$ values measured in the Guaymas OMZ.

The modelled Si isotope composition of the benthic flux is -0.97‰, which is lower than the $\delta^{30}Si$
value of the bottom water (+0.8‰). The higher bottom water $\delta^{30}Si$ value along with the low Si
concentration (~30µM), which is lower than the ambient water column Si concentration (~80µM),
indicates that a certain amount of Si must directly re-precipitate at the sediment water interface. Still
the $\delta^{30}Si$ of the bottom water is lower compared to the ambient water column, showing a benthic Si



flux with low $\delta^{30}$Si values at continental margin settings, which is also in excellent agreement with
previously modelled and calculated $\delta^{30}$Si values (Ehlert et al., 2016; Grasse et al., 2016). These
findings show that benthic Si fluxes at continental margins are a source of low $\delta^{30}$Si values to the
ocean and need to be taken into account in future marine Si budget models.

4.4 Controlling processes and the impact on the global marine Si cycle

Stable and radioactive Si isotope data revealed significant sedimentary import and export processes
influencing the marine silica cycle (Ehlert et al., 2013, 2016; Tréguer and De La Rocha, 2013; Grasse
et al., 2016; Rahman et al., 2017; Sutton et al., 2018). Diatom burial removes about 9.9 Tmol yr$^{-1}$ Si
from the ocean to the sediments, however, effects of terrigenous silicate dissolution and reverse
silicate weathering on $bSiO_2$ burial, preservation, and the benthic Si flux (and its Si isotope
composition) are not well constrained (Sutton et al., 2018). It has previously been shown that silicate
minerals dissolve in deep methanogenic sediments where the dissolution process is favored by high
$CO_2$ and organic ligand concentrations in ambient pore fluids (Wallmann et al., 2008). Similar to
chemical weathering on land, the dissolution of terrigenous silicate phases in marine sediments leads
to a release of cations and the conversion of $CO_2$ into $HCO_3^-$. Moreover, this marine weathering
process provides the dissolved Al that is needed for reverse weathering reactions. Our OMZ data
show for the first time that marine silicate weathering (dissolution of terrigenous silicates) also
occurs in OMZ surface sediments where it can outpace reverse weathering (precipitation of
authigenic silicates). Our study indicates that ambient environmental conditions appear to
significantly influence the balance between marine weathering and reverse weathering and thereby
the Si flux back to the ocean. Pore fluid $\delta^{30}$Si$_{pf}$ values depend on a complex interplay between $bSiO_2$,
terrigenous silicate dissolution, and authigenic aluminosilicate precipitation, however, the controlling
factors that determine which process dominates are difficult to constrain (Fig. 5). In view of the OMZ
settings (Guaymas Basin versus Peruvian margin), the most pronounced difference is the MAR$_{terr}$
which is significantly higher in the Guaymas Basin (252 g m$^{-2}$ yr$^{-1}$; calculated by multiplying the
terrigenous content derived in Eq. (1) with the MAR from Eq. (2)) than at the Peruvian margin (100 g
m$^{-2}$ yr$^{-1}$; MAR from Ehlert et al., 2016; terrigenous content calculated after Eq. (1) with 6 wt% $bSiO_2$,
15 wt% OC, 8 wt% $CaCO_3$) (Fig. 5a, b). The high terrigenous detritus content is supplied via rivers in
the Guaymas Basin (Calvert, 1966; DeMaster, 1981). In combination with the high MAR$_{terr}$ in the
Guaymas OMZ, high water/rock ratios (high porosity) additionally promote dissolution processes (Fig.
5). Lower MAR$_{terr}$ and water/rock ratios found in the Peruvian upwelling margin appear to limit the
dissolution rate of terrigenous phases and promote authigenic aluminosilicate precipitation (Fig. 5b,
c), shifting pore fluid $\delta^{30}$Si$_{pf}$ to higher values compared to the corresponding $\delta^{30}$Si$_{bSiO2}$ and $\delta^{30}$Si$_{bw}$





values. This illustrates that the pore fluid $\delta^{30}Si_{pf}$ values of apparently similar settings (e.g. OMZ sites)
highly depend on the ambient environmental conditions and are not easily transferable.

4.5 Hydrothermal impact on the marine Si cycle

Findings of this study show that additional Si sources like hydrothermal input appear to affect the
oceanic $\delta^{30}Si$ values only in close vicinity to the hydrothermal fields. The $\delta^{30}Si$ values of the
hydrothermal plume (+0.7 to +1.4‰) are highly diluted by seawater (≥ 94%, Table 2) and  thus
deviate from hydrothermal fluid $\delta^{30}Si$ values (-0.3‰; De La Rocha et al., 2000). However, the
currently available data set regarding $\delta^{30}Si$ values of hydrothermal fluids is limited (two data points;
De La Rocha et al., 2000), even though they are in excellent agreement with oceanic crust $\delta^{30}Si$
values (-0.29±; Savage et al., 2010), the rock through which hydrothermal fluids circulate and gain
their Si isotopic signature. In our data set, no correlation exists between the $\delta^{30}Si$ values and the Si
concentration of the hydrothermal plume (Fig. 1S) and instead the $\delta^{30}Si$ values are predominantly
controlled by Si precipitation, likely in the hydrothermal conduit during ascent or after discharge in
contact with colder seawater. Temperature variations and interlinked precipitation rates were found
in addition to co-precipitation with Al or Fe to cause large Si fractionation such that precipitates are
enriched in $^{28}Si$ (Geilert et al., 2014, 2015; Oelze et al., 2015; Roerdink et al., 2015; Zheng et al.,
2016). The varying impacts of these factors can also explain why the diluted hydrothermal plume
$\delta^{30}Si$ values with the highest hydrothermal share (Table 2) does not show the lowest $\delta^{30}Si$ values,
indicative of hydrothermal fluids, given that Si is more reactive compared to Mg, the element on
which the hydrothermal share calculations are based (see supplement from Berndt et al., 2016).  The
large range in hydrothermal plume $\delta^{30}Si$ values, which clearly show high degrees of seawater
dilution, illustrates the complexity of precipitation processes when hydrothermal fluids get in contact
with cold seawater and which requires further investigations especially with respect to the impact on
the global marine Si cycle.

5.   CONCLUSIONS


Marine silicate weathering and reverse weathering impact the pore fluid isotopic composition of
sediments and are key processes of the marine silica cycle. In the Guaymas Basin, these processes
have been studied under markedly differing thermal and redox conditions. Si isotope compositions of
pore fluids combined with those of biogenic silica and ambient bottom waters helped to decipher
marine weathering and reverse weathering reactions, which would have remained undetected by
elemental concentrations alone and highlight the importance of Si isotope studies to constrain early



diagenetic reactions. Si concentrations and $\delta^{30}Si_{pf}$ signatures are the result of the interplay between
silica dissolution and Si precipitation, however, the involved phases differ significantly between the
study sites. Large differences in $\delta^{30}Si_{pf}$ values in a regionally constrained basin show that oxic/anoxic
conditions, hydrothermal fluids, water/rock ratios and the input of terrigenous material strongly
affect the pathways and turnover rates of Si in marine sediments. The light $\delta^{30}Si_{pf}$ and $\delta^{30}Si_{BW}$ values
from the Guaymas OMZ confirm earlier studies suggesting a light Si isotope value of the benthic Si
flux (Ehlert et al., 2016; Grasse et al., 2016), which need to be taken into account in future oceanic
mass balances of Si and in modelling studies concerning the isotopic Si cycle. Environmental settings,
in particular the MARs of terrigenous material, water/rock ratios, and redox conditions appear to be
the major factors controlling the balance between marine silicate weathering and reverse
weathering and the Si isotope fractionation in pore fluids of marine sedimentary settings and need to
be considered particularly in marine Si isotope studies.

AUTHOR CONTRIBUTION

SG, CH, MS, and FS helped sampling and processing of the samples onboard. SG, PG, and KD
conducted the Si isotope measurements. SG, CE, PG, KD, FS, and MF helped interpreting the data.
KW designed the reactive transport model. SG prepared the manuscript with the contribution of all
authors.

COMPETING INTEREST

The authors declare that they have no conflict of interest.

ACKNOWLEDGEMENTS

This work was part of the MAKS project funded by the German Ministry of Science and Education
(BMBF). We appreciate the support of the master and crew of the R/V Sonne during the SO241
cruise. We thank Regina Surberg, Bettina Domeyer, and Anke Bleyer for analytical support during the
cruise and on shore. Further thanks go to Tabitha Riff, Jutta Heinze, and Tyler Goepfert. Additional
support of this work was provided by EU-COST Action ES1301 "FLOWS" (www.flows-cost.eu) and the
German Collaborative Research Centre (SFB) 754: Climate – Biogeochemistry Interactions in the
Tropical Ocean funded by the German Science Foundation.




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






**Tables**

Table 1: Pore fluid Si concentration (µM), $\delta^{30}Si_{pf}$ values (‰) as well as biogenic silica weight fraction
(bSiO$_2$ in wt%), Al/Si ratio (mM/M), $\delta^{30}Si_{bSiO2}$ values (‰), porosity (Ø), Al and K contents (wt%) for the
basin sites, hydrothermal site, and OMZ site.

Table 2: Water column and hydrothermal plume Si concentration (µM) and Si isotope values (‰).
Additionally, the share of hydrothermal fluids within the hydrothermal plume is given based on the
calculation provided by Berndt et al. (2016) in their supplementary materials.

**Figures**

Fig. 1. A) Location map of the sampling stations in the Guaymas Basin, Gulf of California. Black square
in the overview map indicates the sampling area. B) Sedimentary bSiO$_2$ content at each sampling
station. Water column stations were above MUC15-02 (VCTD02) in the basin, the hydrothermal site
(VCTD06, 09), and at the OMZ site (VCTD07).

Fig 2: Depth (cmbsf) profiles for all stations for pore fluid Si concentration (Si(OH)$_4$) in µM (grey
symbols) and $\delta^{30}Si_{pf}$ values (colored symbols) and biogenic opal weight fraction (bSiO$_2$) in wt% (grey
symbols) and $\delta^{30}Si_{bSiO2}$ values (colored symbols). The dashed line is the $\delta^{30}Si$ value of the deep basin
(VCTD02) and the dotted line represents the $\delta^{30}Si$ value of the water column in the OMZ (VCTD07).
The uppermost Si isotope data point in the pore fluid diagrams refers to the bottom water (labelled
BW). Note the different depth scale for the OMZ site. The brackets around the MUC22-04 bottom
water Si concentration value indicate possible surface water contamination. Error bars not indicated
are within symbol size. The long-term error (2SD) of international standards is indicated in the upper
right $\delta^{30}Si_{pf}$-depth profile.

Fig. 3. Pore fluid $\delta^{30}Si$ values are displayed versus the inverse Si concentration (1/Si) for the basin
sites, the hydrothermal site, and the OMZ site. Error bars not indicated are within symbol size. Mixing
curves are calculated after Eq. (3) between the respective water column and the average bSiO$_2$ $\delta^{30}Si$
value for all sites (see text for details).

Fig. 4. Asymptotic Si concentration (a) and the pore fluid $\delta^{30}Si_{pf}$ values (b) as a function of the
terrigenous/bSiO$_2$ ratio for the basin sites, the hydrothermal site, and the OMZ site in the Guaymas





Basin. An exponential increase in silicate concentrations with decreasing terrigenous/$bSiO_2$ ratio is
observed, which is not reflected by corresponding systematic changes in $\delta^{30}Si_{pf}$. The values for the
terrigenous/$bSiO_2$ ratio defining the global trend (grey dots) are from the Southern Ocean, Scotia
Sea, Norwegian Sea, NE Atlantic, Juan de Fuca Ridge, Arabian Sea, and the Peru Basin (Van Cappellen
and Qiu, 1997a; Rabouille et al., 1997; Rickert, 2000).

Fig. 5. Conceptual model of the processes influencing pore fluid $\delta^{30}Si_{pf}$ values in the Guaymas Basin
(a) and the Peruvian margin (b). Bold values in the sediment show the average pore fluid $\delta^{30}Si_{pf}$
values. Arrow length indicates the dominating process (dissolution versus precipitation). The $\delta^{30}Si$
values in the hydrothermal plume indicate dilution with seawater (see section 4.5). (c) The average
pore fluid $\delta^{30}Si_{pf}$ values are shown, indicating the dominance of precipitation or dissolution processes
for the three settings in the Guaymas Basin and the Peruvian OMZ.

Fig. 6. Data and model results for OMZ core. a: Porosity. b: Biogenic opal concentration in solid
phase. c: K/Al ratio in solid phase. d: Dissolved silica concentration in pore fluids. e: Dissolved
potassium in pore fluids. f: Isotopic composition of dissolved silica. g: rate of biogenic opal
dissolution.  h: Rate of authigenic phase precipitation. i: Rate of terrigenous phase dissolution.





Table 1

| Station/MUC# / Station name | Latitude (N) / Longitude (W) | Water depth (m) | Depth (cmbsf) | Pore fluid | | | Sediment | | | | | | |
|---|---|---|---|---|---|---|---|---|---|---|---|---|---|
| | | | | Si (µM) | $\delta^{30}Si$ pf (‰) | 2SD (‰) | $bSiO_2$ (wt%) | (Al/Si) $bSiO_2$ (mM/M) | $\delta^{30}Si$ $bSiO_2$ (‰) | 2SD (‰) | Φ | Al mg g$^{-1}$ | K* mg g$^{-1}$ |
| SO241-33/11/ Basin site | 27° 33.301' 111° 32.883' | 1855 | BW | 173 | 1.9 | 0.2 | - | - | - | - | - | - | - |
| | | | 0.5 | 381 | 1.3 | 0.2 | 22.6 | 39 | 0.8 | 0.1 | 0.932 | 44.4 | 13.7 |
| | | | 1.5 | 455 | - | - | 23.1 | - | - | - | 0.920 | 45.2 | 13.9 |
| | | | 2.5 | 563 | 1.2 | 0.1 | 24.2 | - | - | - | 0.905 | 46.3 | 14.1 |
| | | | 3.5 | 635 | - | - | 22.4 | - | - | - | 0.894 | - | - |
| | | | 4.5 | 685 | - | - | - | - | - | - | 0.892 | 49.1 | 14.7 |
| | | | 6 | 686 | - | - | 25.2 | - | - | - | 0.875 | 39.2 | 12.4 |
| | | | 8 | 745 | - | - | - | - | - | - | 0.857 | - | - |
| | | | 10 | 726 | - | - | 21.9 | - | - | - | 0.852 | - | - |
| | | | 12.5 | 737 | 0.9 | 0.2 | 18.5 | - | - | - | 0.826 | 53.2 | 16.0 |
| | | | 15.5 | 750 | - | - | 14.4 | - | - | - | 0.800 | 61.4 | 17.9 |
| | | | 18.5 | 751 | - | - | 14.8 | - | - | - | 0.787 | - | - |
| | | | 22 | 712 | 1.2 | 0.2 | 19.6 | 26 | 0.8 | 0.2 | 0.801 | 59.5 | 17.5 |
| SO241-22/04/ Basin site | 27° 28.165' 111° 28.347' | 1839 | BW | 54 | 2.0 | 0.2 | - | - | - | - | - | - | - |
| | | | 0.5 | 349 | 1.0 | 0.2 | 11.6 | 37 | 0.9 | 0.2 | 0.910 | 57.8 | 17.0 |
| | | | 1.5 | 377 | - | - | - | - | - | - | 0.907 | - | - |
| | | | 2.5 | 394 | - | - | - | - | - | - | 0.890 | 53.6 | 16.4 |
| | | | 3.5 | 421 | - | - | - | - | - | - | 0.897 | - | - |
| | | | 4.5 | 474 | - | - | - | - | - | - | 0.895 | - | - |
| | | | 5.5 | 558 | 1.1 | 0.1 | 10.9 | - | - | - | 0.893 | 58.7 | 17.3 |
| | | | 7 | 590 | - | - | - | - | - | - | 0.895 | - | - |
| | | | 9 | 637 | 1.2 | 0.2 | 13.1 | - | - | - | 0.890 | 58.6 | 17.2 |
| | | | 11 | 636 | - | - | - | - | - | - | 0.891 | - | - |
| | | | 13 | 597 | - | - | - | - | - | - | 0.895 | - | - |
| | | | 15.5 | 545 | - | - | - | - | - | - | 0.896 | - | - |
| | | | 18.5 | 440 | 1.5 | 0.2 | 7.6 | - | - | - | 0.895 | 57.9 | 17.3 |
| | | | 22 | 404 | - | - | - | - | - | - | 0.876 | - | - |
| | | | 26 | 364 | 1.3 | 0.2 | 9.8 | 71 | 0.5 | 0.2 | 0.842 | 66.6 | 19.4 |





| Site | Depth (m) | Latitude / Longitude | | | | | | | | | | |
|---|---|---|---|---|---|---|---|---|---|---|---|---|
| SO241-23/05/ Basin site | 1726 | 27° 30.282' 111° 40.770' | BW | 216 | 1.8 | 0.3 | - | - | - | - | - | - | - |
| | | | 0.5 | 485 | 1.4 | 0.2 | 28.2 | 23 | 0.9 | 0.2 | 0.934 | 30.3 | 9.2 |
| | | | 1.5 | 610 | - | - | - | - | - | - | 0.921 | - | - |
| | | | 2.5 | 713 | - | - | - | - | - | - | 0.921 | - | - |
| | | | 3.5 | 753 | 1.3 | 0.1 | 30.1 | - | - | - | 0.923 | 31.8 | 9.8 |
| | | | 4.5 | 751 | - | - | - | - | - | - | 0.918 | - | - |
| | | | 5.5 | 726 | - | - | - | - | - | - | 0.908 | 33.1 | 9.8 |
| | | | 7 | 713 | - | - | - | - | - | - | 0.904 | - | - |
| | | | 9 | 731 | 1.1 | 0.2 | 28.3 | - | - | - | 0.894 | 35.3 | 10.6 |
| | | | 11 | 690 | - | - | - | - | - | - | 0.879 | - | - |
| | | | 13 | 651 | - | - | - | - | - | - | 0.829 | - | - |
| | | | 15.5 | 663 | - | - | - | - | - | - | 0.829 | 27.6 | 8.0 |
| | | | 18.5 | 664 | - | - | - | - | - | - | 0.825 | - | - |
| | | | 22 | 640 | 1.3 | 0.4 | 21.4 | 15 | 0.4 | 0.1 | 0.807 | 27.7 | 8.1 |
| SO241-15/02/ Basin site | 1845 | 27°26.925' 111°29.926' | BW | 178 | 1.6 | 0.2 | - | - | - | - | - | - | - |
| | | | 0.5 | 474 | 1.2 | 0.3 | 26.7 | 19 | 0.9 | 0.3 | 0.958 | 31.5 | 10.7 |
| | | | 1.5 | 569 | - | - | 24.7 | - | - | - | 0.943 | 33.9 | 10.8 |
| | | | 2.5 | 590 | - | - | 27.3 | - | - | - | 0.935 | 35.7 | 11.1 |
| | | | 3.5 | 610 | - | - | 27.7 | - | - | - | 0.918 | 35.9 | 11.0 |
| | | | 4.5 | 605 | - | - | - | - | - | - | 0.927 | 30.4 | 9.2 |
| | | | 5.5 | 607 | 1.2 | 0.2 | 30.4 | - | - | - | 0.924 | 37.2 | 11.3 |
| | | | 7 | 635 | - | - | - | - | - | - | 0.908 | 37.2 | 11.4 |
| | | | 9 | 711 | - | - | 25.2 | - | - | - | 0.913 | 34.5 | 10.6 |
| | | | 11 | 746 | - | - | 30.1 | - | - | - | 0.918 | 32.0 | 9.9 |
| | | | 13 | 673 | - | - | - | - | - | - | 0.920 | 32.2 | 10.2 |
| | | | 15.5 | 707 | 1.2 | 0.2 | 47.6 | - | - | - | 0.927 | 23.9 | 7.7 |
| | | | 18.5 | 727 | - | - | - | - | - | - | 0.929 | 21.3 | 6.6 |
| | | | 22 | 753 | - | - | - | - | - | - | 0.919 | 29.4 | 9.1 |
| | | | 26 | 737 | - | - | - | - | - | - | 0.882 | 46.4 | 13.9 |
| | | | 30 | 781 | 1.2 | 0.2 | 23.6 | 57 | 1.0 | 0.2 | 0.864 | 51.8 | 15.4 |
| SO241-66/16/ Hydrothermal site | 1842 | 27° 24.577' 111° 23.265' | BW | 254 | 1.5 | 0.2 | - | - | - | - | - | - | - |
| | | | 0.5 | 427 | 2.0 | 0.2 | 13.3 | 52 | 0.8 | 0.2 | 0.911 | 17.4 | 4.8 |
| | | | 1.5 | 554 | - | - | - | - | - | - | 0.900 | 18.9 | 5.2 |



| Station | Lat | Lon | Water depth | Depth | Density | | | | | | | $\Phi$ | | |
|---|---|---|---|---|---|---|---|---|---|---|---|---|---|---|
| | | | | 2.5 | 702 | - | - | - | - | - | - | 0.855 | 17.4 | 4.4 |
| | | | | 3.5 | 752 | - | - | - | - | - | - | 0.820 | 16.7 | 4.1 |
| | | | | 4.5 | 781 | - | - | - | - | - | - | 0.808 | 16.0 | 3.8 |
| | | | | 5.5 | 781 | 2.2 | 0.1 | 11.0 | - | - | - | 0.793 | 15.4 | 3.5 |
| | | | | 6.5 | 875 | - | - | - | - | - | - | 0.775 | 14.8 | 3.3 |
| | | | | 9 | 877 | - | - | - | - | - | - | 0.770 | 13.4 | 2.7 |
| | | | | 11 | 892 | - | - | 7.0 | - | - | - | 0.742 | 9.4 | 1.3 |
| | | | | 13 | 914 | - | - | 4.7 | - | - | - | 0.810 | 17.8 | 3.9 |
| | | | | 15.5 | 903 | - | - | 14.6 | - | - | - | 0.802 | 21.9 | 5.4 |
| | | | | 18.5 | 888 | 1.8 | 0.1 | 8.2 | 35 | 0.9 | 0.1 | 0.620 | 2.9 | - |
| SO241-29/09/ OMZ site | 27°42.410' | 111°13.656' | 665 | BW | 31 | 0.8 | 0.2 | - | - | - | - | | - | - |
| | | | | 0.5 | 247 | -0.5 | 0.3 | 15.3 | - | - | - | 0.970 | 35.1 | 12.67 |
| | | | | 1.5 | 425 | - | - | - | - | - | - | 0.954 | 46.6 | 14.0 |
| | | | | 2.5 | 458 | -0.3 | 0.1 | 16.9 | - | - | - | 0.945 | 48.9 | 14.9 |
| | | | | 3.5 | 492 | - | - | - | - | - | - | 0.947 | 47.2 | 14.7 |
| | | | | 4.5 | 569 | - | - | - | - | - | - | 0.939 | 48.6 | 15.0 |
| | | | | 5.5 | 618 | 0.3 | 0.2 | 15.3 | - | - | - | 0.935 | 51.1 | 15.3 |
| | | | | 6.5 | 655 | - | - | - | - | - | - | 0.928 | 50.4 | 15.4 |
| | | | | 7.5 | 735 | - | - | - | - | - | - | 0.923 | 53.4 | 16.2 |
| | | | | 9 | 763 | 0.0 | 0.2 | 13.5 | - | - | - | 0.926 | 51.8 | 15.8 |
| | | | | 11 | 754 | - | - | - | - | - | - | 0.926 | 39.5 | 10.1 |
| | | | | 13 | 753 | - | - | - | - | - | - | 0.929 | 48.6 | 14.2 |
| | | | | 15 | 767 | - | - | 16.23 | 26 | 0.8 | 0.1 | 0.909 | 53.3 | 15.9 |
| | | | | 18.5 | 772 | - | - | - | - | - | - | 0.913 | 53.8 | 15.9 |
| | | | | 20.5 | 781 | -0.2 | 0.2 | 18.1 | - | - | - | 0.911 | 53.3 | 15.7 |
| | | | | 23.5 | 763 | - | - | - | - | - | - | 0.906 | 56.1 | 16.8 |
| | | | | 26.5 | 765 | - | - | - | - | - | - | 0.919 | 47.8 | 14.5 |
| | | | | 29 | 763 | - | - | - | - | - | - | 0.929 | 41.5 | 12.6 |
| | | | | 30 | 768 | - | - | - | - | - | - | 0.925 | 45.9 | 13.9 |
| | | | | 38 | 760 | 0.8 | 0.2 | 21.9 | - | - | - | 0.925 | 45.8 | 13.5 |

$\Phi$ = porosity
* porosity corrected





Table 2

| Cruise-Station/VCTD#/ bottle#/Station name | Latitude (N) / Longitude (W) | Depth | Si | $\delta^{30}$Si | 2SD | hydrothermal fluid share* |
|---|---|---|---|---|---|---|
| | | (mbsl) | (µM) | (‰) | (‰) | (%) |
| **Water column** | | | | | | |
| SO241-12/02/ | 27° 26.133 | 1844 | 163 | 1.5 | 0.1 | 0.1 |
| Basin site | 111° 30.268 | | | | | |
| SO241-42/07/ | 27° 42.411 | 586 | 78 | 1.5 | 0.2 | 0 |
| OMZ site | 111° 13.663 | | | | | |
| **Hydrothermal plume** | | | | | | |
| SO241-67/09/06/ | 27° 24.750 | 1800 | 253 | 0.7 | 0.1 | 2.1 |
| Hydrothermal site | 111° 23.240 | | | | | |
| SO241-67/09/09/ | 27° 24.750 | 1800 | 206 | 1.4 | 0.2 | 0.2 |
| Hydrothermal site | 111° 23.240 | | | | | |
| SO241-67/09/12/ | 27° 24.750 | 1800 | 690 | 1.0 | 0.2 | 5.7 |
| Hydrothermal site | 111° 23.240 | | | | | |

* calculation in Berndt et al. (2016)





Fig. 1

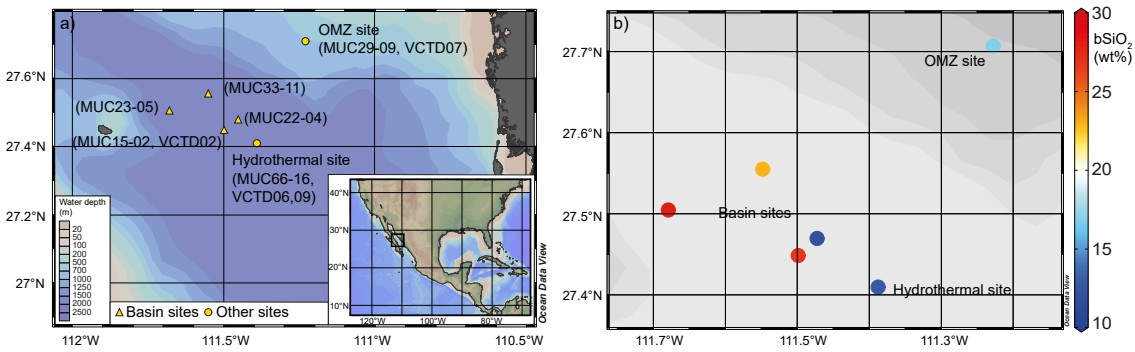



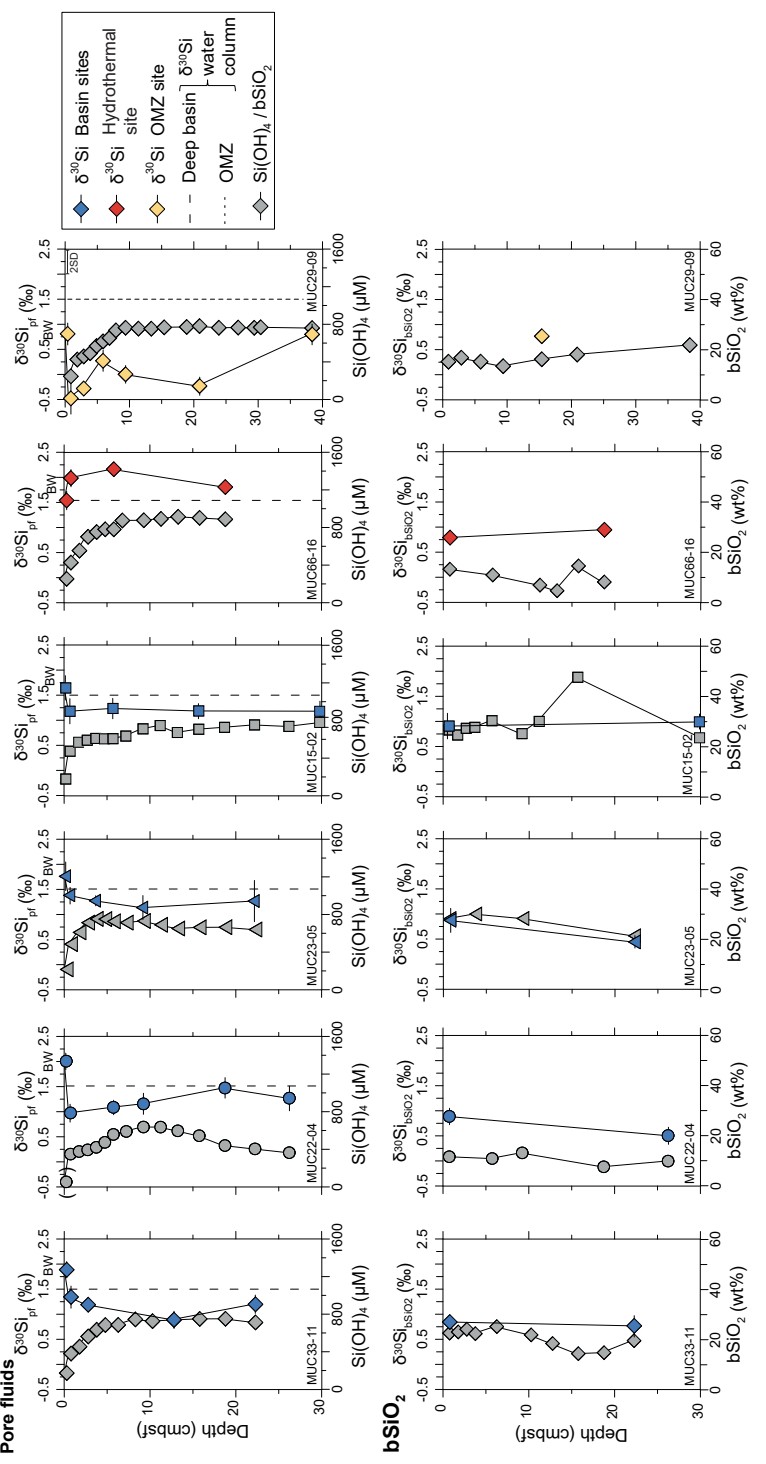

Fig. 2





Fig. 3

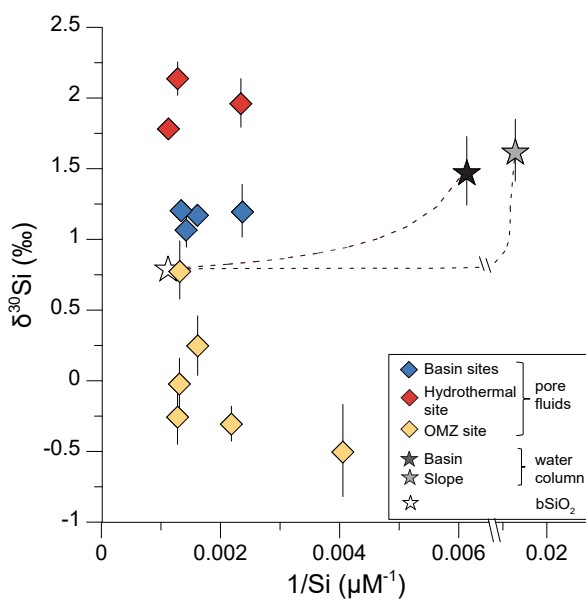



Fig. 4

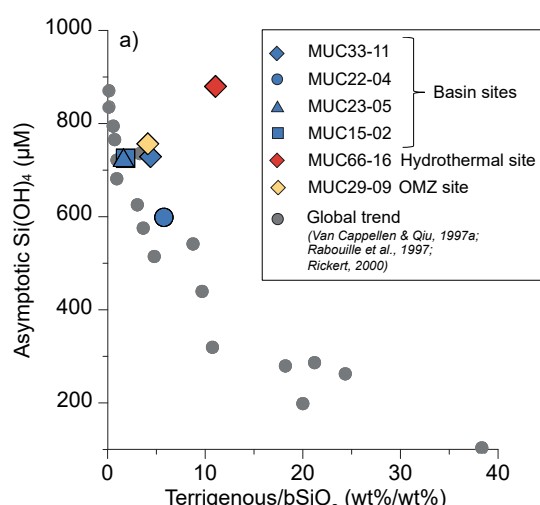
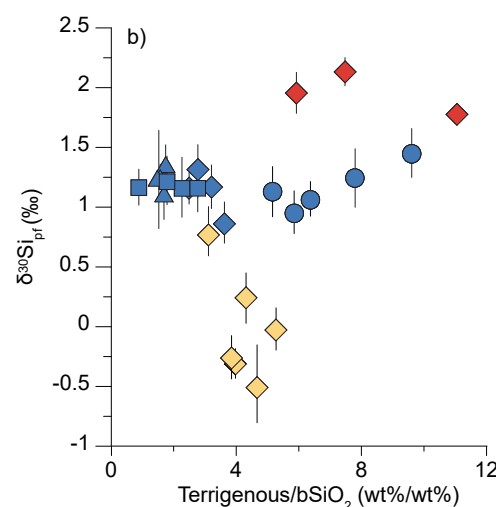



Fig. 5

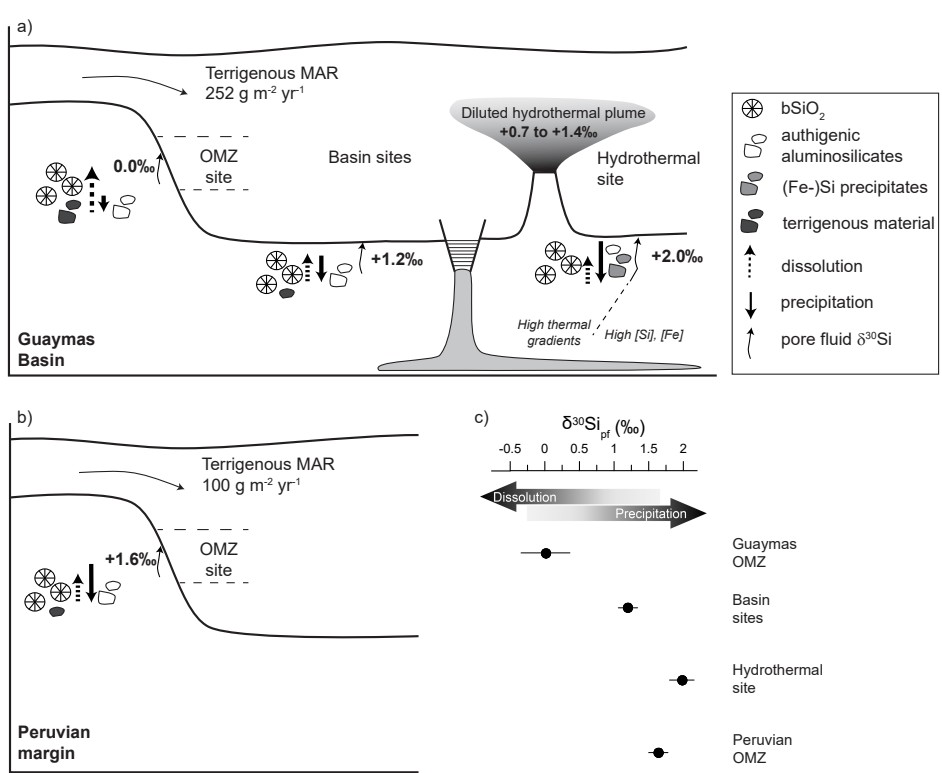



Fig. 6

