# Peer review of "Impact of ambient conditions on the Si isotope fractionation in marine pore fluids during early diagenesis"

_Biogeosciences, 2019_

## Referee Comment (RC1) · Anonymous Referee #1 · 14 Jan 2020

This paper presents new pore water and sediments data from the Guaymas Basin in the Pacific, focusing on silicon and stable silicon isotopes, early diagenetic processes, and implications for silicon cycling in the oceans. The authors present new and high-quality data, adding to a relatively sparse literature on the subject, and explore their interpretation with a model. The paper is very well-written and enjoyable to read. I have only a few comments and suggestions for where the methods and discussion could be expanded. As such, I am fully supportive of the publication of this manuscript with minor revisions.

1) I would like to see some more detail in the methods and supplementary information.

[Figure]

Firstly, on page 6, line 182, the authors describe drying down the dissolved bSiO2 samples prior to analysis. Could there have been any problems with loss of Si at this stage? Could the authors comment upon this and perhaps include yield data?

Secondly, I think that it would be incomplete not to mention the possibility of isotopic fractionation during dissolution of biogenic opal in section 4.1. I appreciate that this fractionation is poorly constrained, with very few studies that do not agree (Demarest et al., 2009; Egan et al., 2012, Wetzel et al., 2014). As such, I think that it's acceptable to say that we can assume that there is no appreciable fractionation, but the possibility should be included as a caveat.

Thirdly, I would like to see more information about the modelling in the supplementary information. Such models are highly sensitive to the assumed dissolution rates of the involved phases. If any other group wanted to reconstruct this model, it would be challenging to do so without knowing exactly how e.g. the terrigenous phase dissolution rate profile parameter was quantified. Could the authors please include the actual equations used, linking depth in the sediment column with kinetic constants (i.e. the equations used to produce Figure S2)?

I would like to know more about the sensitivity of the model to the assumed values of K/Al for the different phases. In particular, what is the sensitivity of the outputs to the ratio for the authigenic phase? It seems that the assumed value is for sediments from a very different environmental setting – can the authors justify the use of values from the Gulf of Mexico for modelling the Guaymas Basin? How does precipitation at a hydrothermal site impact this ratio (section 4.3.2.)? I would suggest that the authors include a sensitivity experiment, perhaps with a few different profile plots for different (reasonable) assumed K/Al values, in the supplementary information. It might also be interesting to investigate the sensitivity of the model to variations in other 'constants' too, especially those that are poorly constrained or found to be variable in natural systems (e.g. the solubility of biogenic opal). Lastly, the caveats of the model are buried in the supplementary information, and I would like to see them more integrated into main

text.

2) I also think that there are aspects of the discussion that could be expanded upon to utilize the full range of data available.

Firstly, the XRD data is not referred to at all the discussion. How does the clay mineralogy inform on the discussion? Does it help with constraining reverse weathering reactions and/or, for example, the potential shifts in K/Al within the sediments (e.g. section 4.3.3.)?

Secondly, I also do not think that the pore water trace metals are used to their full potential. For example, are there any trends in the [Fe] data from the hydrothermal site to suggest that Fe-cycling could be impacting silicon isotope fractionation? (section 4.3.3.)

3) Other minor comments:

I'd suggest that the authors should be consistent and use either "pore fluids" or "pore waters" throughout the text.

Line 32: Please change "the only other marine setting where Si isotopes have been investigated to constrain early diagenetic processes" to "the only other OMZ marine setting where Si isotopes have been investigated to constrain early diagenetic processes", to acknowledge that other marine settings have been investigated (e.g. Ng et al., 2020).

Additional references:

Demarest, M. S., Brzezinski, M. A., & Beucher, C. P. (2009). Fractionation of silicon isotopes during biogenic silica dissolution. Geochimica et Cosmochimica Acta, 73(19), 5572-5583

Egan, K. E., Rickaby, R. E., Leng, M. J., Hendry, K. R., Hermoso, M., Sloane, H. J., ... & Halliday, A. N. (2012). Diatom silicon isotopes as a proxy for silicic acid utilisation: a

Southern Ocean core top calibration. Geochimica et Cosmochimica Acta, 96, 174-192

Wetzel, F., De Souza, G. F., & Reynolds, B. C. (2014). What controls silicon isotope fractionation during dissolution of diatom opal?. Geochimica et Cosmochimica Acta, 131, 128-137

---

## Referee Comment (RC2) · Jill Sutton (Referee) · 29 Jan 2020

The manuscript by Geilert and co-authors presents stable silicon isotope data obtained from pore fluid, water column, hydrothermal plume, and sediment samples at several different sites in the Guaymas Basin. The authors present the geochemical differences observed at these different sites and discuss the implications of processes, such as early diagenesis, on the cycling of Si in the marine environment. The data are of high quality and the authors have done an excellent job at interpreting and presenting their data. The manuscript is timely and worthy of publication in Biogeosciences. This is the second time that I have reviewed this manuscript (first time was with GCA), and I have

very few additional suggestions to make, and only of an editorial manner.

Lines 13 and 44 (for example) – Can the authors please check their usage of the definition of Si throughout the text? Here the authors describe silica as Si (line 13) but also silicon as Si. I would prefer that the authors of the manuscript use Si to describe silicon and $SiO_2$ for silica (consistent with the other abbreviations used throughout the text).

Line 226 – please change "insignificant" to "unimportant".

Throughout – it is De La Rocha (and not de la Rocha). Varela (2004) should be Varela et al. 2004

---

## Referee Comment (RC3) · Damien Cardinal (Referee) · 13 Feb 2020

This manuscript presents new interesting silicon isotope data from sediment pore fluids (d30Sipf) in different marine environments from Guyamas Basin and Gulf of California. The data are of high quality and include d30Sipf from sites located into Oxygen Minimum Zone or under the influence of hydrothermal vents both of which are particularly new in terms of Si isotopic system. This work is well written and certainly deserves publication in Biogeosciences after improvement / clarification of some parts of the discussion. I have compiled my concerns under the scope of the three main concerns below, which do not necessarily imply much work to be implemented.

[Figure]

1) The role of bioturbation is never mentioned in the main text (e.g. 76-77) despite it is an important process in the upper cm (cf. e.g. Rabouille et al. 1997 for Si). It is however a parameter of the model. However, the model is used only for the OMZ site where we could expect null or negligible bioturbation because it is very low oxygen environments. How could bioturbation not affect the porefluid profiles of the other sites? In contrast why this has been considered at the OMZ site? Some discussion should be added in the main text.

2) Identification of minerals under or over saturated in the pf is not sufficiently directly addressed in the discussion. See e.g.:

- 457-459. It is unclear how "Si concentrations lower than amorphous Si and quartz solubilities, indicating precipitation of the respective mineral phase during pore fluid ascent". If the DSipf is lower than equilibrium concentration, then Si should rather dissolve than precipitate.

- What are the minerals which are actually oversaturated in pf ? Why saturation indexes of primary and secondary minerals have not been calculated e.g. using data of Table 1S?

- 480-481 and 499-512. Could dissolution of primary minerals (e.g. feldspars) supply DSi in pf? If not, justify why this can be ruled out. Even though primary minerals are less prone to dissolution than clays, they are likely to be undersaturated in the pf and may have some impact on DSi and d30Sipf.

- Why there is no mineral data on the OMZ core (Table 2S) despite this is the site for which the discussion is the most developed?

- Fig.3. What is the uncertainty we have on Si concentration for dissolved bSi to build the mixing line that has been taken at equilibrium (900 uM)? The equilbrium concentration is theoretical however why would the dissolution of bSiO2 be at equilibrium and give Sipf at 900 uM for the end-member chosen in Fig. 3?

3) Model set up from line 524 and in the Supplement. The sensitivity of the model to its main hypotheses is not sufficiently discussed in the main text and there is a lack of justification for some of its core parameters.

- The average value of clay used in the model is -2 pmil and reference to Frings et al. (2016) is given for this. However, average clays in Frings et al. is not at -2 pmil. I'm not sure it is actually calculated, but from the figure, it should be more between -1.5 and -1 pmil. This would be also consistent with the review of Sutton et al. (2018) In which the world average value of secondary minerals is at -1.08 pmil. Similarly, in Bayon et al. (2018) the average clay d30Si from river sediment fluctuate from -1.5 to -0.32 pmil depending on climatic regimes. How does it affect model outputs when using a more realistic d30Si of clay (i.e. -1.5 or -1 pmil) and/or propagate the uncertainty of this value? In any case, the use of -2 pmil as average clay value is not properly justified.

- Similarly uncertainty on the -2 pmil for the isotopic fractionation during precipitation of authigenic clay should be discussed and taken into account

- Role of bioturbation in the model for the OMZ (cf. first comment)

- Could the model be applied to the other sites, e.g. basin?

4) Minor comments

Throughout the ms, better use heavier / lighter than higher / lower when reference is made to isotopic composition.

176 : Âń The bSiO2 samples were stored in Milli-Q water Âż Does it mean that once separated by Morley et al. (2004) method, the bSiO2 samples were kept in water ? For how long? Dissolution could have occurred with some isotopic fractionation?

238. Typo, 2 times Âń and Âż

301 Is it worth keeping MUC-22-04 whose bottom SW has been contaminated ? (likely by surface SW)

366: the kinetics of reverse weathering is poorly known, especially in situ and is certainly not immediate. So do not to use such wording Âń as soon as Si is released (...), it reprecipitates Âż. Moreover not all Si reprecipitates, otherwise there won't be more DSi in pore fluids than in bottom water (indeed in their previous work, Elhert et al. 2016 have quantified that only 24% of dissolved bSi reprecipitates). This sentence needs to be corrected.

371-372 From the three references cited here, only Elhert et al. 2016 has estimated fractionation factors for reverse weathering, the other two refer to continental weathering (Georg et al., 2006 and Opfegerlt et al., 2013). Remove them or specify it since this sentence is misleading.

395-398. Sentence unclear / grammatically incorrect

397. At least one reference should be cited for Si isotope fractionation during bSi dissolution (e.g. DeMarest et al., 2009).

440-441. This sentence is too affirmative given the level of discussion at this stage of the ms. It could be changed to e.g. "Thus, at Basin sites both K/Al ratios of sediments and the heavier d30Sipf are in agreement to recognise bSi dissolution followed by authigenic clay formation as significant processes taking place".

527: typo in isotope

Table S5 Typo Âń auf Âż

Fig. 5. It should be mentioned in the caption that Fig. 5b is from another study (Ehlert et al. 2016?)

Fig. 6. Need to define red and black dashed line in the caption without having to go through the text in the ms.

Please also note the supplement to this comment:
https://www.biogeosciences-discuss.net/bg-2019-481/bg-2019-481-RC3-

supplement.pdf

---

## Author Comment (AC1) · 25 Feb 2020

This paper presents new pore water and sediments data from the Guaymas Basin in the Pacific, focusing on silicon and stable silicon isotopes, early diagenetic processes, and implications for silicon cycling in the oceans. The authors present new and high quality data, adding to a relatively sparse literature on the subject, and explore their interpretation with a model. The paper is very well-written and enjoyable to read. I have only a few comments and suggestions for where the methods and discussion could be expanded. As such, I am fully supportive of the publication of this manuscript with minor revisions.

[Figure]

Dear Reviewer, Thank you very much for your positive feedback and the appreciation of our work. We mainly agree with the comments and suggestions you have raised and we will incorporate the requested changes in the manuscript, in case of a positive evaluation by the editor. Please find below our answers to your comments.

I would like to see some more detail in the methods and supplementary information.

1) Firstly, on page 6, line 182, the authors describe drying down the dissolved bSiO2 samples prior to analysis. Could there have been any problems with loss of Si at this stage? Could the authors comment upon this and perhaps include yield data?

2) The described process of bSiO2 digestion is a standard procedure following Reynolds et al. (2008) and Ehlert et al. (2012). Drying of the samples was shown by Ehlert et al. (2012) to have no effect on the Si isotopic composition of the samples. Additionally, Si is not volatile during evaporation and therefore fractionation affects not likely to occur.

3) We will comment on this and include the method references in the method section of the main text.

1) Secondly, I think that it would be incomplete not to mention the possibility of isotopic fractionation during dissolution of biogenic opal in section 4.1. I appreciate that this fractionation is poorly constrained, with very few studies that do not agree (Demarest et al., 2009; Egan et al., 2012, Wetzel et al., 2014). As such, I think that it's acceptable to say that we can assume that there is no appreciable fractionation, but the possibility should be included as a caveat.

2) As the referee mentions, we exclude significant effects on pore fluid $\delta 30Si_{pf}$ values, given the highly unconstrained and diverging results of former studies (Demarest et al., 2009; Egan et al., 2012; Wetzel et al., 2014). Nevertheless, the discussion of possible Si isotope fractionation in dependence of bSiO2 dissolution is an important aspect.

3) We will add a short paragraph in section 4.1 to address this caveat.

[Figure]

1) Thirdly, I would like to see more information about the modelling in the supplementary information. Such models are highly sensitive to the assumed dissolution rates of the involved phases. If any other group wanted to reconstruct this model, it would be challenging to do so without knowing exactly how e.g. the terrigenous phase dissolution rate profile parameter was quantified. Could the authors please include the actual equations used, linking depth in the sediment column with kinetic constants (i.e. the equations used to produce Figure S2)? I would like to know more about the sensitivity of the model to the assumed values of K/Al for the different phases. In particular, what is the sensitivity of the outputs to the ratio for the authigenic phase? It seems that the assumed value is for sediments from a very different environmental setting – can the authors justify the use of values from the Gulf of Mexico for modelling the Guaymas Basin? How does precipitation at a hydrothermal site impact this ratio (section 4.3.2.)? I would suggest that the authors include a sensitivity experiment, perhaps with a few different profile plots for different (reasonable) assumed K/Al values, in the supplementary information. It might also be interesting to investigate the sensitivity of the model to variations in other 'constants' too, especially those that are poorly constrained or found to be variable in natural systems (e.g. the solubility of biogenic opal). Lastly, the caveats of the model are buried in the supplementary information, and I would like to see them more integrated into main text.

2) The assumed K/Al ratio is taken from sediments in the Amazon River delta, which are considered as the end product of reverse weathering reactions, given the complete conversion of the diatom frustule to authigenic aluminosilicates (Michalopoulos et al., 2000). The complete conversion of the diatom frustule is due to the input of highly reactive terrigenous minerals in the Amazon deltaic setting (Michalopoulos and Aller, 2004). The state of conversion of the diatom assemblage in the Guaymas Basin is difficult to assess, but similar K/Al ratios compared to the Amazon setting indicate a comparable high maturity state. At the hydrothermal site, the K/Al ratio is similar to the basin sites and indicates a similar state of conversion of the diatom frustule. Lower K/Al ratios and with that a lower maturity state of the diatom frustule were found in

experiments by Loucaides et al. (2010). We agree that sensitivity tests will benefit the modelling outcomes and interpretation. For the modelling itself not K/Al but K/Si ratios from the same literature were used (see supplement) and modelling outcomes were recalculated for K/Al ratios. Therefore we conducted sensitivity tests with varying K/Si ratios and also sensitivity tests concerning the solubility constant of biogenic opal. The sensitivity tests showed that lower K/Si ratios (and with that lower K/Al ratios) could not reproduce the measured K/Al ratios in the OMZ. Therefore, we conclude that the assumed K/Al ratios of Michalopoulos et al. (2000) are valid also for authigenic minerals in the Guaymas OMZ.

3) We will incorporate the sensitivity tests and the outcomes in the supplement. Also, we will move the caveats of the model, which are currently discussed in the supplement, to the main text. Additional sensitivity tests regarding the solubility of biogenic opal, the $\delta 30Si$ values of the dissolving terrigenous phase and the Si isotope fractionation during authigenic clay formation were conducted following also the recommendations by reviewer #3.

I also think that there are aspects of the discussion that could be expanded upon to utilize the full range of data available.

1) Firstly, the XRD data is not referred to at all the discussion. How does the clay mineralogy inform on the discussion? Does it help with constraining reverse weathering reactions and/or, for example, the potential shifts in K/Al within the sediments (e.g. section 4.3.3.)?

2) In section 4.3.2 of the original manuscript we actually used the XRD data on amorphous SiO2 in the discussion of hydrothermal processes (lines 459-465). However, we made no attempt to use XRD data for the detection of reverse weathering reactions because it is difficult to distinguish authigenic clays formed during these reactions from terrigenous clays that are very abundant in our study area. For this reason, we did in fact not analyze the OMZ sediments with XRD so that we are unfortunately not able to

follow this reviewer recommendation.

1) Secondly, I also do not think that the pore water trace metals are used to their full potential. For example, are there any trends in the [Fe] data from the hydrothermal site to suggest that Fe-cycling could be impacting silicon isotope fractionation? (section 4.3.3.)

2) Unfortunately, the available data set is too scarce to identify possible Si isotope fractionation induced by Fe. Experimental results by Zheng et al. (2016) indicate that $\delta30$Sipf values should increase with the Fe/Si ratio in the solids and pore fluids. We see a similar trend in our data from the basin sites and hydrothermal site indicating a possible Fe-induced fractionation (see figure below). Data from the OMZ site, in contrast, deviate from this apparent trend (however, only based on two data points) and show lower $\delta30$Sipf values as what would be expected regarding the Fe/Si ratio. This could be related to the likely one step fractionation during Fe-Si co-precipitation at the hydrothermal site and the existence of multiple Fe redox cycles inducing Si dissolution and re-precipitation and with that multiple fractionation steps at the OMZ site (see original manuscript lines 483-497). Furthermore, any Fe-induced Si isotope fractionation is likely superimposed by the dissolution of terrigenous clays as shown by the reactive transport model. Natural Fe-induced Si isotope fractionation needs further investigation in future studies in order to be able to identify magnitudes of fractionation if other fractionating processes take place simultaneously.

3) We will add the figure below and a short discussion on Fe-induced fractionation to the supplementary information in order to address the reviewer's comment.

Other minor comments:

1) I'd suggest that the authors should be consistent and use either "pore fluids" or "pore waters" throughout the text.

3) We will use "pore fluids" throughout the text.

[Figure]

1) Line 32: Please change "the only other marine setting where Si isotopes have been investigated to constrain early diagenetic processes" to "the only other OMZ marine setting where Si isotopes have been investigated to constrain early diagenetic processes", to acknowledge that other marine settings have been investigated (e.g. Ng et al., 2020).

3) We will add 'OMZ' to the sentence and refer to the previous work conducted in other marine settings (e.g. Ng et al., 2020).

References

Loucaides S., Michalopoulos P., Presti M., Koning E., Behrends T. and Van Cappellen P. (2010) Seawater-mediated interactions between diatomaceous silica and terrigenous sediments: Results from long-term incubation experiments. Chem. Geol. 270, 68–79. Available at: http://dx.doi.org/10.1016/j.chemgeo.2009.11.006.

Michalopoulos P. and Aller R. C. (2004) Early diagenesis of biogenic silica in the Amazon delta: Alteration, authigenic clay formation, and storage. Geochim. Cosmochim. Acta 68, 1061–1085.

Michalopoulos P., Aller R. C. and Reeder R. J. (2000) Conversion of diatoms to clays during early diagenesis in tropical, continental shell muds. Geology 28, 1095–1098.

Zheng X., Beard B. L., Reddy T. R., Roden E. E. and Johnson C. M. (2016) Abiologic silicon isotope fractionation between aqueous Si and Fe ( III ) -Si gel in simulated Archean seawater : Implications for Si isotope records in Precambrian sedimentary rocks. Geochemica Cosmochim. Acta 187, 102–122.

[Figure]

[Figure]

**Fig. 1.** Additional figure to reviewer comment 2-2. Si isotope fractionation in dependence of the Fe/Si ratio in the fluids. Experimental data from Zheng et al. (2016) are shown for comparison.

---

## Author Comment (AC2) · 25 Feb 2020

The manuscript by Geilert and co-authors presents stable silicon isotope data obtained from pore fluid, water column, hydrothermal plume, and sediment samples at several different sites in the Guaymas Basin. The authors present the geochemical differences observed at these different sites and discuss the implications of processes, such as early diagenesis, on the cycling of Si in the marine environment. The data are of high quality and the authors have done an excellent job at interpreting and presenting their data. The manuscript is timely and worthy of publication in Biogeosciences. This is the second time that I have reviewed this manuscript (first time was with GCA), and I have

very few additional suggestions to make, and only of an editorial manner.

Dear Jill Sutton, Thank you very much for your positive evaluation and the valuable feedback and comments on our manuscript. We will incorporate the changes you propose in case of a positive evaluation by the editor.

1) Lines 13 and 44 (for example) – Can the authors please check their usage of the definition of Si throughout the text? Here the authors describe silica as Si (line 13) but also silicon as Si. I would prefer that the authors of the manuscript use Si to describe silicon and SiO2 for silica (consistent with the other abbreviations used throughout the text).

2+3) We will use the abbreviation Si for silicon and SiO2 for silica throughout the revised text.

1) Line 226 – please change "insignificant" to "unimportant".

2+3) We will change the word accordingly.

1) Throughout – it is De La Rocha (and not de la Rocha). Varela (2004) should be Varela et al. 2004

2+3) We will change the references accordingly.

---

## Author Comment (AC3) · 25 Feb 2020

This manuscript presents new interesting silicon isotope data from sediment pore fluids (d30Sipf) in different marine environments from Guyamas Basin and Gulf of California. The data are of high quality and include d30Sipf from sites located into Oxygen Minimum Zone or under the influence of hydrothermal vents both of which are particularly new in terms of Si isotopic system. This work certainly deserves publication in Biogeosciences after improvement / clarification of some parts of the discussion. I have compiled my concerns under the scope of the three main concerns below, which do not necessarily imply much work to be implemented.

[Figure]

Dear Damien Cardinal, Thank you very much for your positive feedback and valuable comments to our manuscript. We are happy to discuss and incorporate the aspects you raised below in a revised version of this manuscript in case of a positive evaluation by the editor.

1) The role of bioturbation is never mentioned in the main text (e.g. 76-77) despite it is an important process in the upper cm (cf. e.g. Rabouille et al. 1997 for Si). It is however a parameter of the model. However, the model is used only for the OMZ site where we could expect null or negligible bioturbation because it is very low oxygen environments. How could bioturbation not affect the porefluid profiles of the other sites? In contrast why this has been considered at the OMZ site? Some discussion should be added in the main text.

2) We agree that bioturbation and bioirrigation and with that the mixing of upper sediments and pore fluids are important processes which were not mentioned in the main text of the manuscript. Bioturbation is expected to have the largest impact within the upper ~10 cm below seafloor (cmbsf) and bioirrigation within the upper ~20 cmbsf. However, the homogeneity of the pore fluid $\delta 30 Si_{pf}$ values with depth (see original manuscript lines 421-423) suggests that the effects of bioturbation/bioirrigation on pore fluid composition are largely compensated by fast reactions inducing rapid isotope exchange. Unfortunately, we have no independent data to quantify rates of bioturbation and bioirrigation in our study area. Therefore, we limited the modelling to the OMZ site where the biogenic mixing proceeds at very low rates due to the absence of large benthic biota.

3) Nevertheless, we will briefly discuss the potential effects of bioturbation/bioirrigation at the hydrothermal and basin sites in section 4.2.

Identification of minerals under or over saturated in the pf is not sufficiently directly addressed in the discussion. See e.g.:

1) 457-459. It is unclear how "Si concentrations lower than amorphous Si and quartz

solubilities, indicating precipitation of the respective mineral phase during pore fluid ascent". If the DSipf is lower than equilibrium concentration, then Si should rather dissolve than precipitate.

2+3) The sentence will be rephrased according to calculated saturation indices (see comment below).

1) What are the minerals which are actually oversaturated in pf? Why saturation indexes of primary and secondary minerals have not been calculated e.g. using data of Table 1S?

2) We will calculate saturation indices for pure silica phases (amorphous SiO2, quartz) using the program and databases of PHREEQC (Parkhurst and Appelo, 1999). However, no saturation indices can be calculated for the most important precipitating phase discussed in this manuscript, namely authigenic aluminosilicates because we have no data on the dissolved Al concentrations and in-situ pH values of the pore fluids. For the hydrothermal site, saturation indices show that quartz is supersaturated and amorphous silica is close to saturation. Consequently, quartz can precipitate directly from pore fluids at present. However, due to the dynamics of hydrothermal systems, this can be subject to changes and supersaturation of amorphous silica is likely to be obtained occasionally, due to the ascent of Si enriched fluids from greater depth as indicated by the presence of amorphous silica cement in the hydrothermally affected sediments (see lines 459-462 in the original manuscript).

3) We will add a paragraph discussing saturation indices in section 4.3.2.

1) 480-481 and 499-512. Could dissolution of primary minerals (e.g. feldspars) supply DSi in pf? If not, justify why this can be ruled out. Even though primary minerals are less prone to dissolution than clays, they are likely to be undersaturated in the pf and may have some impact on DSi and d30Sipf.

2) It is true that reactive primary minerals like feldspars may dissolve in the pore fluids

(e.g. Singer, 1980; Wilson, 2004). However, the $\delta 30Si$ values of feldspars are rather high compared to clays with an average of -0.17‰ (Georg et al., 2009; Savage et al., 2011) and their dissolution alone cannot be responsible for the shift to low $\delta 30Sipf$ values in the OMZ (see also answer to comment #3 Modelling). During changing environmental conditions immature clays like e.g. chlorite can transform and dissolve (Singer, 1980), often accelerated by organic ligands and iron reduction in the clay structure (Anderson and Raiswell, 2004; and original manuscript lines 502-508). We agree, that primary mineral dissolution is likely to take place, however, superimposed by clay dissolution, shifting OMZ pore fluid $\delta 30Sipf$ to the observed low values. We conducted a sensitivity test considering feldspar dissolution (see also answer to comment #3). Our results indicate that feldspar dissolution alone cannot create the low $\delta 30Sipf$ values in the OMZ.

3) We will add the results of this sensitivity test to the supplement and address the results and a discussion on primary silicate dissolution in the main text.

1) Why there is no mineral data on the OMZ core (Table 2S) despite this is the site for which the discussion is the most developed?

2) We agree that XRD data for the OMZ site would be helpful and could add another argument for clay dissolution and/ or authigenic mineral precipitation. Unfortunately, by the time when the XRD analyses were conducted, the focus of the study was not on the OMZ site. Only in the course of the manuscript writing and handling, the focus shifted to the OMZ site. However, we also think that the recognition of authigenic mineral formation is rather difficult to decipher based on XRD data (see comment to reviewer #1, point 2-1) and would not have supported the discussion to a large degree because the clay mineralogy of the OMZ site is probably dominated by riverine clays that are very abundant in our study area and complicate the detection of authigenic clay formation. We show, that we can fully explain the observed low $\delta 30Sipf$ values by clay dissolution. The interpretation is additionally supported by our modeling and K/Al data.

1) Fig.3. What is the uncertainty we have on Si concentration for dissolved bSi to build the mixing line that has been taken at equilibrium (900 uM)? The equilbrium concentration is theoretical however why would the dissolution of bSiO2 be at equilibrium and give Sipf at 900 uM for the end-member chosen in Fig. 3?

2) The assumed bSiO2 equilibrium concentration of $900\mu$M is an experimentally determined value for siliceous sediment (Van Cappellen and Qiu, 1997; see also lines 390-392). This concentration value takes into account early diagenetic reactions like the incorporation of Al in the diatom frustule. This diagenesis-affected concentration value is lower than equilibrium concentrations of acid-cleaned bSiO2 (see Van Cappellen and Qiu, 1997 and references therein). We agree, however, that equilibrium concentrations might vary from site to site depending on the maturity of the diatom frustules.

3) Therefore we will add an uncertainty of $\pm$ $150\mu$M Si to the assumed concentration (c.f. Van Cappellen and Qiu, 1997) and add a range in Fig. 3 and a comment in the caption. Note that the uncertainty of the equilibrium solubility of bSiO2 has only minor impact on the calculated mixing curves.

Model set up from line 524 and in the Supplement. The sensitivity of the model to its main hypotheses is not sufficiently discussed in the main text and there is a lack of justification for some of its core parameters.

1) The average value of clay used in the model is -2 pmil and reference to Frings et al. (2016) is given for this. However, average clays in Frings et al. is not at -2 pmil. I'm not sure it is actually calculated, but from the figure, it should be more between -1.5 and -1 pmil. This would be also consistent with the review of Sutton et al. (2018) In which the world average value of secondary minerals is at -1.08 pmil. Similarly, in Bayon et al. (2018) the average clay d30Si from river sediment fluctuate from -1.5 to -0.32 pmil depending on climatic regimes. How does it affect model outputs when using a more realistic d30Si of clay (i.e. -1.5 or -1 pmil) and/or propagate the uncertainty of this

value? In any case, the use of -2 pmil as average clay value is not properly justified.

2) The terrigenous clays brought to the basin by river discharge are likely phyllosilicates like kaolinites (e.g. Georg et al., 2006; Frings et al., 2014) which are associated with lower $\delta$30Si values caused by the larger Si isotope fractionation factor associated with single layer phyllosilicates (Opfergelt et al., 2012). Therefore, it is valid to assume a clay $\delta$30Si value of -2‰.However, we agree that the reference to Frings et al. (2016) is not sufficiently explaining this assumption and we agree that a sensitive test of the model taking into account various clay $\delta$30Si values will improve the manuscript and the significance of the model results. Therefore, we conducted sensitivity tests for clay $\delta$30Si values covering a range of -2 to -1 ‰ and for primary mineral dissolution with $\delta$30Si values close to zero. Results of the sensitivity tests show that dissolution of terrigenous material with higher $\delta$30Si values than -2‰ cannot reproduce the measured $\delta$30Si values in the OMZ pore fluids. Only if the fractionation factor is lowered to -1‰ terrigenous material with $\delta$30Si values of -1.7‰ can produce the observed values (see also comment below). In conclusion, the dissolving terrigenous phase is strongly depleted in 30Si and only clay dissolution can produce the low pore fluid $\delta$30Si values in the OMZ.

3) We will add these sensitivity tests to the supplement and refer to the results in the main text.

1) Similarly uncertainty on the -2 pmil for the isotopic fractionation during precipitation of authigenic clay should be discussed and taken into account

2) We agree that the model will benefit from sensitivity tests concerning the Si isotope fractionation factor. We conducted sensitivity tests applying $\Delta$30Si values of -1‰ and 0‰ following Opfergelt et al. (2012). A fractionation factor of -1 ‰ reproduces the measured pore fluid $\delta$30Si values in the OMZ if a terrigenous phase with slightly higher $\delta$30Si values (-1.7‰ compared to -2‰ dissolves.

3) We will include the outcomes of this sensitivity test in the supplement and refer to

the results in the main text.

1) Role of bioturbation in the model for the OMZ (cf. first comment)

2) Bioturbation is of minor importance for the OMZ given the absence of large benthic biota under anoxic conditions. The model incorporates a small bioturbation coefficient. However, the depth of the bioturbated layer is limited to 1 cm considering the absence of burrowing organisms.

1) Could the model be applied to the other sites, e.g. basin?

2) In contrast to the OMZ, the other sites are likely influenced by bioturbation and bioirrigation even if the impact on pore fluid $\delta$30Sipf values appears to be compensated by fast reactions (see answer to comment #1). Unfortunately, we have no independent data to quantify rates of bioturbation and bioirrigation in our study area. Therefore, we limited the modelling to the OMZ site where the biogenic mixing proceeds at very low rates due to the absence of large benthic biota.

Minor comments

1) Throughout the ms, better use heavier / lighter than higher / lower when reference is made to isotopic composition.

2) In the original manuscript we used the expression higher and lower when referring to $\delta$30Si values, as a value by its nature cannot be light or heavy. We used the expression lighter and heavier when referring to isotopic compositions (e.g. in lines 28 and 402 in the original manuscript).

1) 176 : Âń The bSiO2 samples were stored in Milli-Q water Âż Does it mean that once separated by Morley et al. (2004) method, the bSiO2 samples were kept in water ? For how long? Dissolution could have occurred with some isotopic fractionation?

2) The cleaned diatom samples were stored in MQ-water for several days. Dissolution of bSiO2 is unlikely given that the pH of the MQ-water ($\sim$ 5) is not favoring bSiO2

dissolution. Dissolution rates increase quickly between pH 9 to 10.7 (Iler, 1979) and that is also the reason why digestion of the bSiO2 samples is conducted in an alkaline medium. Additionally, the water-bSiO2 mixture was transferred to a Teflon vial for further handling, so in the unlikely case of fractionation effects during dissolution, the bulk would have been further processed and no isotopic signal was lost.

1) 238. Typo, 2 times Âń and Âż

2+3) We will remove the second 'and'.

1) 301 Is it worth keeping MUC-22-04 whose bottom SW has been contaminated? (likely by surface SW)

2) Because of completeness of the results section, we prefer to leave the sample MUC-22-04 in and report the Si concentration and $\delta$30Si values.

1) 366: the kinetics of reverse weathering is poorly known, especially in situ and is certainly not immediate. So do not to use such wording Âń as soon as Si is released (...), it reprecipitates Âż. Moreover not all Si reprecipitates, otherwise there won't be more DSi in pore fluids than in bottom water (indeed in their previous work, Elhert et al. 2016 have quantified that only 24% of dissolved bSi reprecipitates). This sentence needs to be corrected.

3) We will correct the sentence accordingly.

1) 371-372 From the three references cited here, only Elhert et al. 2016 has estimated fractionation factors for reverse weathering, the other two refer to continental weathering (Georg et al., 2006 and Opfegerlt et al., 2013). Remove them or specify it since this sentence is misleading.

3) We will remove the references Georg et al., 2009 and Opfergelt et al., 2013.

1) 395-398. Sentence unclear / grammatically incorrect

3) We will rewrite the sentence.

1) 397. At least one reference should be cited for Si isotope fractionation during bSi dissolution (e.g. DeMarest et al., 2009).

2+3) We will discuss the potential impact on Si isotope fractionation during dissolution and add the following references: Demarest et al., 2009, Egan et al., 2012, Wetzel et al., 2014 (see also comment to reviewer #1).

1) 440-441. This sentence is too affirmative given the level of discussion at this stage of the ms. It could be changed to e.g. "Thus, at Basin sites both K/Al ratios of sediments and the heavier d30Sipf are in agreement to recognise bSi dissolution followed by authigenic clay formation as significant processes taking place".

3) We will rephrase the sentence accordingly.

1) 527: typo in isotope

3) We will correct the typo.

1) Table S5 Typo Âń auf Âż

3) We will correct the typo.

1) Fig. 5. It should be mentioned in the caption that Fig. 5b is from another study (Ehlert et al. 2016?)

3) We will include the reference Ehlert et al., 2016 in the caption of Fig. 5.

1) Fig. 6. Need to define red and black dashed line in the caption without having to go through the text in the ms.

3) We will include the definitions of the red and black dashed lines in the caption.

References

Anderson T. F. and Raiswell R. (2004) SOURCES AND MECHANISMS FOR THE ENRICHMENT OF HIGHLY REACTIVE IRON IN EUXINIC BLACK SEA SEDIMENTS. Am. J. Sci. 304, 203–233.

Van Cappellen P. and Qiu L. Q. (1997) Biogenic silica dissolution in sediments of the Southern Ocean.1. Solubility. Deep. Res. Part Ii-Topical Stud. Oceanogr. 44, 1109–1128.

Frings P. J., Clymans W., Fontorbe G., De La Rocha C. L. and Conley D. J. (2016) The continental Si cycle and its impact on the ocean Si isotope budget. Chem. Geol. 425, 12–36. Available at: http://dx.doi.org/10.1016/j.chemgeo.2016.01.020.

Frings P. J., Rocha C. D. La, Struyf E., Pelt D. Van, Schoelynck J., Hudson M. M., Gondwe M. J., Wolski P., Mosimane K., Gray W., Schaller J. and Conley D. J. (2014) Tracing silicon cycling in the Okavango Delta, a sub-tropical flood-pulse wetland using silicon isotopes. Geochim. Cosmochim. Acta 142, 132–148. Available at: http://www.sciencedirect.com/science/article/pii/S0016703714004694.

Georg R. B., Reynolds B. C., Frank M. and Halliday a. N. (2006) Mechanisms controlling the silicon isotopic compositions of river waters. Earth Planet. Sci. Lett. 249, 290–306.

Georg R. B., Zhu C., Reynolds B. C. and Halliday A. N. (2009) Stable silicon isotopes of groundwater, feldspars, and clay coatings in the Navajo Sandstone aquifer, Black Mesa, Arizona, USA. Geochim. Cosmochim. Acta 73, 2229–2241. Available at: http://dx.doi.org/10.1016/j.gca.2009.02.005.

Iler R. K. (1979) The Chemistry of Silica., John Wiley & Sons Inc, New York.

Parkhurst B. D. L. and Appelo C. a J. (1999) User's Guide To PHREEQC (version 2) — a Computer Program for Speciation, and Inverse Geochemical Calculations. Exch. Organ. Behav. Teach. J. D, 326. Available at: http://downloads.openchannelsoftware.org/PHREEQC/manual.pdf.

Savage P. S., Georg R. B., Williams H. M., Burton K. W. and Halliday A. N. (2011) Silicon isotope fractionation during magmatic differentiation. Geochim. Cosmochim. Acta 75, 6124–6139.

Singer A. (1980) The Paleoclimatic Interpretation of Clay Minerals in Soils and Weathering Profiles. Earth-Science Rev. 15, 303–326.

Wilson M. J. (2004) Weathering of the primary rock-forming minerals: processes, products and rates. Clay Miner. 39, 233–266.
* * *

---

## Author Response (AR1)

This paper presents new pore water and sediments data from the Guaymas Basin in the Pacific, focusing on silicon and stable silicon isotopes, early diagenetic processes, and implications for silicon cycling in the oceans. The authors present new and high quality data, adding to a relatively sparse literature on the subject, and explore their interpretation with a model. The paper is very well-written and enjoyable to read. I have only a few comments and suggestions for where the methods and discussion could be expanded. As such, I am fully supportive of the publication of this manuscript with minor revisions.

Dear Reviewer,
Thank you very much for your positive feedback and the appreciation of our work. We mainly agree with the comments and suggestions you have raised and we will incorporate the requested changes in the manuscript, in case of a positive evaluation by the editor. Please find below our answers to your comments. All line numbers refer to the revised manuscript.

I would like to see some more detail in the methods and supplementary information.

1) Firstly, on page 6, line 182, the authors describe drying down the dissolved $bSiO_2$ samples prior to analysis. Could there have been any problems with loss of Si at this stage? Could the authors comment upon this and perhaps include yield data?

2) The described process of $bSiO_2$ digestion is a standard procedure following Reynolds et al. (2008) and Ehlert et al. (2012). Drying of the samples was shown by Ehlert et al. (2012) to have no effect on the Si isotopic composition of the samples. Additionally, Si is not volatile during evaporation and therefore fractionation affects not likely to occur.

3) The references and the comment on drying effects on the Si isotope composition were added in lines 179-181.

1) Secondly, I think that it would be incomplete not to mention the possibility of isotopic fractionation during dissolution of biogenic opal in section 4.1. I appreciate that this fractionation is poorly constrained, with very few studies that do not agree (Demarest et al., 2009; Egan et al., 2012, Wetzel et al., 2014). As such, I think that it's acceptable to say that we can assume that there is no appreciable fractionation, but the possibility should be included as a caveat.

2) As the referee mentions, we exclude significant effects on pore fluid $\delta^{30}Si_{pf}$ values, given the highly unconstrained and diverging results of former studies (Demarest et al., 2009; Egan et al., 2012; Wetzel et al., 2014). Nevertheless, the discussion of possible Si isotope fractionation in dependence of $bSiO_2$ dissolution is an important aspect.

3) We added a short paragraph in section 4.1 (lines 397-401) to address this caveat.

1) Thirdly, I would like to see more information about the modelling in the supplementary information. Such models are highly sensitive to the assumed dissolution rates of the involved phases. If any other group wanted to reconstruct this model, it would be challenging to do so without knowing exactly how e.g. the terrigenous phase dissolution rate profile parameter was quantified. Could the authors please include the actual equations used, linking depth in the sediment column with kinetic constants (i.e. the equations used to produce Figure S2)? I would like to know more about the sensitivity of the model to the assumed values of K/Al for the different phases. In particular, what is the sensitivity of the outputs to the ratio for the authigenic phase? It seems that the assumed value is for sediments from a very different environmental setting – can the authors justify the use of values from the Gulf of Mexico for modelling the Guaymas Basin? How does precipitation at a hydrothermal site impact this ratio (section 4.3.2.)? I would suggest that the authors include a sensitivity experiment, perhaps with a few different profile plots for different (reasonable) assumed K/Al values, in the supplementary information. It might also be interesting to investigate the sensitivity of the model to variations in other 'constants' too, especially those that are poorly constrained or found to be variable in natural systems (e.g. the solubility of biogenic opal). Lastly, the caveats of the model are buried in the supplementary information, and I would like to see them more integrated into main text.

2) The assumed K/Al ratio is taken from sediments in the Amazon River delta, which are considered as the end product of reverse weathering reactions, given the complete conversion of the diatom frustule to authigenic aluminosilicates (Michalopoulos et al., 2000). The complete conversion of the diatom frustule is due to the input of highly reactive terrigenous minerals in the Amazon deltaic setting (Michalopoulos and Aller, 2004). The state of conversion of the diatom assemblage in the Guaymas Basin is difficult to assess, but similar K/Al ratios compared to the Amazon setting indicate a comparable high maturity state. At the hydrothermal site, the K/Al ratio is similar to the basin sites and indicates a similar state of conversion of the diatom frustule. Lower K/Al ratios and with that a lower maturity state of the diatom frustule were found in experiments by Loucaides et al. (2010). We agree that sensitivity tests will benefit the modelling outcomes and interpretation. For the modelling itself not K/Al but K/Si ratios from the same literature were used (see supplement) and modelling outcomes were recalculated for K/Al ratios. Therefore we conducted sensitivity tests with varying K/Si ratios and also sensitivity tests concerning the solubility constant of biogenic opal. The sensitivity tests showed that lower K/Si ratios (and with that lower K/Al ratios) could not reproduce the measured K/Al ratios in the OMZ. Therefore, we conclude that the assumed K/Al ratios of Michalopoulos et al. (2000) are valid also for authigenic minerals in the Guaymas OMZ.

3) Sensitivity tests and the outcomes were added in the supplement. Also, the caveats of the model were moved to the main text in lines 550-553.

Additional sensitivity tests regarding the solubility of biogenic opal, the $\delta^{30}$Si values of the dissolving terrigenous phase and the Si isotope fractionation during authigenic clay formation were conducted following also the recommendations by reviewer #3.

2) I also think that there are aspects of the discussion that could be expanded upon to utilize the full range of data available.

1) Firstly, the XRD data is not referred to at all the discussion. How does the clay mineralogy inform on the discussion? Does it help with constraining reverse weathering reactions and/or, for example, the potential shifts in K/Al within the sediments (e.g. section 4.3.3.)?

2) In section 4.3.2 of the original manuscript we actually used the XRD data on amorphous $SiO_2$ in the discussion of hydrothermal processes (lines 474-476). However, we made no attempt to use XRD data for the detection of reverse weathering reactions because it is difficult to distinguish authigenic clays formed during these reactions from terrigenous clays that are very abundant in our study area. For this reason, we did in fact not analyze the OMZ sediments with XRD so that we are unfortunately not able to follow this reviewer recommendation.

1) Secondly, I also do not think that the pore water trace metals are used to their full potential. For example, are there any trends in the [Fe] data from the hydrothermal site to suggest that Fe-cycling could be impacting silicon isotope fractionation? (section 4.3.3.)

2) Unfortunately, the available data set is too scarce to identify possible Si isotope fractionation induced by Fe. Experimental results by Zheng et al. (2016) indicate that $\delta^{30}$Si values should increase with the Fe/Si ratio in the solids and pore fluids. We see a similar trend in our data from the basin sites and hydrothermal site indicating a possible Fe-induced fractionation (see figure below). Data from the OMZ site, in contrast, deviate from this apparent trend (however, only based on two data points) and show lower $\delta^{30}$Si$_{pf}$ values as what would be expected regarding the Fe/Si ratio. This could be related to the likely one step fractionation during Fe-Si co-precipitation at the hydrothermal site and the existence of multiple Fe redox cycles inducing Si dissolution and re-precipitation and with that multiple fractionation steps at the OMZ site (lines 507-510). Furthermore, any Fe-induced Si isotope fractionation is likely superimposed by the dissolution of terrigenous clays as shown by the reactive transport model. Natural Fe-induced Si isotope fractionation needs further investigation in future studies in order to be able to identify magnitudes of fractionation if other fractionating processes take place simultaneously.

3) We added the figure below and a short discussion on Fe-induced fractionation to the supplementary information in order to address the reviewer's comment (reference in the main text in line 510).

[Figure]

*Additional figure to reviewer comment 2-2. Si isotope fractionation in dependence of the Fe/Si ratio in the fluids. Experimental data from Zheng et al. (2016) are shown for comparison.*

3) Other minor comments:

1) I'd suggest that the authors should be consistent and use either "pore fluids" or "pore waters" throughout the text.

2+3) We used "pore fluids" throughout the text.

1) Line 32: Please change "the only other marine setting where Si isotopes have been investigated to constrain early diagenetic processes" to "the only other OMZ marine setting where Si isotopes have been investigated to constrain early diagenetic processes", to acknowledge that other marine settings have been investigated (e.g. Ng et al., 2020).

2+3) We added 'OMZ' to the sentence (line 32) and previous studies conducted in other marine settings are discussed in lines 89-96.

**Referee #2 , Jill Sutton**

The manuscript by Geilert and co-authors presents stable silicon isotope data obtained from pore fluid, water column, hydrothermal plume, and sediment samples at several different sites in the Guaymas Basin. The authors present the geochemical differences observed at these different sites and discuss the implications of processes, such as early diagenesis, on the cycling of Si in the marine environment. The data are of high quality and the authors have done an excellent job at interpreting and presenting their data. The manuscript is timely and worthy of publication in Biogeosciences. This is the second time that I have reviewed this manuscript (first time was with GCA), and I have very few additional suggestions to make, and only of an editorial manner.

Dear Jill Sutton,
Thank you very much for your positive evaluation and the valuable feedback and comments on our manuscript. We will incorporate the changes you propose in case of a positive evaluation by the editor. All line numbers refer to the revised manuscript.

1) Lines 13 and 44 (for example) – Can the authors please check their usage of the definition of Si throughout the text? Here the authors describe silica as Si (line 13) but also silicon as Si. I would prefer that the authors of the manuscript use Si to describe silicon and $SiO_2$ for silica (consistent with the other abbreviations used throughout the text).

2+3) We used the abbreviation Si for silicon and $SiO_2$ for silica throughout the revised text.

1) Line 226 – please change "insignificant" to "unimportant".

2+3) We changed the word accordingly (line 229).

1) Throughout – it is De La Rocha (and not de la Rocha). Varela (2004) should be Varela et al. 2004

2+3) We changed the references accordingly.

**Referee #3, Damien Cardinal**

This manuscript presents new interesting silicon isotope data from sediment pore fluids (d30Sipf) in different marine environments from Guyamas Basin and Gulf of California. The data are of high quality and include d30Sipf from sites located into Oxygen Minimum Zone or under the influence of hydrothermal vents both of which are particularly new in terms of Si isotopic system. This work certainly deserves publication in Biogeosciences after improvement / clarification of some parts of the discussion. I have compiled my concerns under the scope of the three main concerns below, which do not necessarily imply much work to be implemented.

Dear Damien Cardinal,
Thank you very much for your positive feedback and valuable comments to our manuscript. We are happy to discuss and incorporate the aspects you raised below in a revised version of this manuscript in case of a positive evaluation by the editor. All line numbers refer to the revised manuscript.

1) The role of **bioturbation** is never mentioned in the main text (e.g. 76-77) despite it is an important process in the upper cm (cf. e.g. Rabouille et al. 1997 for Si). It is however a parameter of the model. However, the model is used only for the OMZ site where we could expect null or negligible bioturbation because it is very low oxygen environments. How could bioturbation not affect the porefluid profiles of the other sites? In contrast why this has been considered at the OMZ site? Some discussion should be added in the main text.

2) We agree that bioturbation and bioirrigation and with that the mixing of upper sediments and pore fluids are important processes which were not mentioned in the main text of the manuscript. Bioturbation is expected to have the largest impact within the upper ~10 cm below seafloor (cmbsf) and bioirrigation within the upper ~20 cmbsf. However, the homogeneity of the pore fluid $\delta^{30}Si_{pf}$ values with depth (see lines 427-429) suggests that the effects of bioturbation/bioirrigation on pore fluid composition are largely compensated by fast reactions inducing rapid isotope exchange. Unfortunately, we have no independent data to quantify rates of bioturbation and bioirrigation in our study area. Therefore, we limited the modelling to the OMZ site where the biogenic mixing proceeds at very low rates due to the absence of large benthic biota.

3) Nevertheless, we will briefly discuss the potential effects of bioturbation/bioirrigation at the hydrothermal and basin sites in section 4.2 (429-433).

**Identification of minerals under or over saturated** in the pf is not sufficiently directly addressed in the discussion. See e.g.:

1) 457-459. It is unclear how "*Si concentrations lower than amorphous Si and quartz solubilities, indicating precipitation of the respective mineral phase during pore fluid ascent*". If the DSipf is lower than equilibrium concentration, then Si should rather dissolve than precipitate.

2+3) The sentence was rephrased according to calculated saturation indices (see comment below).

1) What are the minerals which are actually oversaturated in pf? Why saturation indexes of primary and secondary minerals have not been calculated e.g. using data of Table 1S?

2) We calculated saturation indices for pure silica phases (amorphous $SiO_2$, quartz) using the program and databases of PHREEQC (Parkhurst and Appelo, 1999). However, no saturation indices can be calculated for the most important precipitating phase discussed in this manuscript, namely authigenic aluminosilicates because we have no data on the dissolved Al concentrations and in-situ pH values of the pore fluids. For the hydrothermal site, saturation indices show that quartz is supersaturated and amorphous silica is close to saturation. Consequently, quartz can precipitate directly from pore fluids at present. However, due to the dynamics of hydrothermal systems, this can be subject to changes and supersaturation of amorphous silica is likely to be obtained occasionally, due to the ascent of Si enriched fluids from greater depth as indicated by the presence of amorphous silica cement in the hydrothermally affected sediments.

3) We added a paragraph discussing saturation indices in section 4.3.2, lines 469-473.

1) 480-481 and 499-512. Could dissolution of primary minerals (e.g. feldspars) supply DSi in pf? If not, justify why this can be ruled out. Even though primary minerals are less prone to dissolution than clays, they are likely to be undersaturated in the pf and may have some impact on DSi and d30Sipf.

2) It is true that reactive primary minerals like feldspars may dissolve in the pore fluids (e.g. Singer, 1980; Wilson, 2004). However, the $\delta^{30}Si$ values of feldspars are rather high compared to clays with an average of -0.17‰ (Georg et al., 2009; Savage et al., 2011) and their dissolution alone cannot be responsible for the shift to low $\delta^{30}Si_{pf}$ values in the OMZ (see also answer to comment #3 Modelling). During changing environmental conditions immature clays like e.g. chlorite can transform and dissolve (Singer, 1980), often accelerated by organic ligands and iron reduction in the clay structure (Anderson and Raiswell, 2004; and lines 522-531). We agree, that primary mineral dissolution is likely to take place, however, superimposed by clay dissolution, shifting OMZ pore fluid $\delta^{30}Si_{pf}$ to the observed low values. We conducted a sensitivity test considering feldspar dissolution (see also answer to comment #3). Our results indicate that feldspar dissolution alone cannot create the low $\delta^{30}Si_{pf}$ values in the OMZ.

3) We added the results of this sensitivity test to the supplement and addressed the results and a discussion on primary silicate dissolution in the main text lines 517-520.

1) Why there is no mineral data on the OMZ core (Table 2S) despite this is the site for which the discussion is the most developed?

2) We agree that XRD data for the OMZ site would be helpful and could add another argument for clay dissolution and/ or authigenic mineral precipitation. Unfortunately, by the time when the XRD analyses were conducted, the focus of the study was not on the OMZ site. Only in the course of the manuscript writing and handling, the focus shifted to the OMZ site. However, we also think that the recognition of authigenic mineral formation is rather difficult to decipher based on XRD data (see comment to reviewer #1, point 2-1) and would not have supported the discussion to a large degree because the clay mineralogy of the OMZ site is probably dominated by riverine clays that are very abundant in our study area and complicate the detection of authigenic clay formation. We show, that we can fully explain the observed low $\delta^{30}Si_{pf}$ values by clay dissolution. The interpretation is additionally supported by our modeling and K/Al data.

1) Fig.3. What is the uncertainty we have on Si concentration for dissolved bSi to build the mixing line that has been taken at equilibrium (900 uM)? The equilbrium concentration is theoretical however why would the dissolution of bSiO2 be at equilibrium and give Sipf at 900 uM for the end-member chosen in Fig. 3?

2) The assumed $bSiO_2$ equilibrium concentration of 900µM is an experimentally determined value for siliceous sediment (Van Cappellen and Qiu, 1997; see also lines 392-394). This concentration value takes into account early diagenetic reactions like the incorporation of Al in the diatom frustule. This diagenesis-affected concentration value is lower than equilibrium concentrations of acid-cleaned $bSiO_2$ (see Van Cappellen and Qiu, 1997 and references therein). We agree, however, that equilibrium concentrations might vary from site to site depending on the maturity of the diatom frustules.

3)

3) Therefore we added an uncertainty of ± 150µM Si to the assumed concentration (c.f. Van Cappellen and Qiu, 1997) and added a range in Fig. 3 and a comment in the caption. Note that the uncertainty of the equilibrium solubility of $bSiO_2$ has only minor impact on the calculated mixing curves.

**3) Model set up** from line 524 and in the Supplement. The sensitivity of the model to its main hypotheses is not sufficiently discussed in the main text and there is a lack of justification for some of its core parameters.

1) The average value of clay used in the model is -2 pmil and reference to Frings et al. (2016) is given for this. However, average clays in Frings et al. is not at -2 pmil. I'm not sure it is actually calculated, but from the figure, it should be more between -1.5 and -1 pmil. This would be also consistent with the review of Sutton et al. (2018) In which the world average value of secondary minerals is at -1.08 pmil. Similarly, in Bayon et al. (2018) the average clay d30Si from river sediment fluctuate from -1.5 to -0.32 pmil depending on climatic regimes. How does it affect model outputs when using a more realistic d30Si of clay (i.e. -1.5 or -1 pmil) and/or propagate the uncertainty of this value? In any case, the use of -2 pmil as average clay value is not properly justified.

2) The terrigenous clays brought to the basin by river discharge are likely phyllosilicates like kaolinites (e.g. Georg et al., 2006; Frings et al., 2014) which are associated with lower $\delta^{30}Si$ values caused by the larger Si isotope fractionation factor associated with single layer phyllosilicates (Opfergelt et al., 2012). Therefore, it is valid to assume a clay $\delta^{30}Si$ value of -2‰. However, we agree that the reference to Frings et al. (2016) is not sufficiently explaining this assumption and we agree that a sensitive test of the model taking into account various clay $\delta^{30}Si$ values will improve the manuscript and the significance of the model results. Therefore, we conducted sensitivity tests for clay $\delta^{30}Si$ values covering a range of -2 to -1 ‰ and for primary mineral dissolution with $\delta^{30}Si$ values close to zero. Results of the sensitivity tests show that dissolution of terrigenous material with higher $\delta^{30}Si$ values than -2‰ cannot reproduce the measured $\delta^{30}Si$ values in the OMZ pore fluids. Only if the fractionation factor is lowered to -1‰, terrigenous material with $\delta^{30}Si$ values of -1.7‰ can produce the observed values (see also comment below). In conclusion, the dissolving terrigenous phase is strongly depleted in $^{30}Si$ and only clay dissolution can produce the low pore fluid $\delta^{30}Si$ values in the OMZ.

3) We added these sensitivity tests to the supplement and refer to the results in the main text (lines 557-559 and 574-578).

1) Similarly uncertainty on the -2 pmil for the isotopic fractionation during precipitation of authigenic clay should be discussed and taken into account

2) We agree that the model will benefit from sensitivity tests concerning the Si isotope fractionation factor. We conducted sensitivity tests applying $\Delta^{30}Si$ values of -1‰ and 0‰ following Opfergelt et al. (2012). A fractionation factor of -1 ‰ reproduces the measured pore fluid $\delta^{30}Si$ values in the OMZ if a terrigenous phase with slightly higher $\delta^{30}Si$ values (-1.7‰ compared to -2‰) dissolves.

3) We included the outcomes of this sensitivity test in the supplement and refer to the results in the main text (lines 557-559 and 574-578).

1) Role of bioturbation in the model for the OMZ (cf. first comment)

2) Bioturbation is of minor importance for the OMZ given the absence of large benthic biota under anoxic conditions. The model incorporates a small bioturbation coefficient. However, the depth of the bioturbated layer is limited to 1 cm considering the absence of burrowing organisms.

1) Could the model be applied to the other sites, e.g. basin?

2) In contrast to the OMZ, the other sites are likely influenced by bioturbation and bioirrigation even if the impact on pore fluid $\delta^{30}Si_{pf}$ values appears to be compensated by fast reactions (see answer to comment #1). Unfortunately, we have no independent data to quantify rates of bioturbation and bioirrigation in our study area. Therefore, we limited the modelling to the OMZ site where the biogenic mixing proceeds at very low rates due to the absence of large benthic biota.

**Minor comments**

1) Throughout the ms, better use heavier / lighter than higher / lower when reference is made to isotopic composition.

2) In the original manuscript we used the expression higher and lower when referring to $\delta^{30}Si$ values, as a value by its nature cannot be light or heavy. We used the expression lighter and heavier when referring to isotopic compositions (e.g. in line 404).

1) 176 : « *The bSiO2 samples were stored in Milli-Q water* » Does it mean that once separated by Morley et al. (2004) method, the bSiO2 samples were kept in water ? For how long? Dissolution could have occurred with some isotopic fractionation?

2) The cleaned diatom samples were stored in MQ-water for several days. Dissolution of $bSiO_2$ is unlikely given that the pH of the MQ-water (~ 5) is not favoring $bSiO_2$ dissolution. Dissolution rates increase quickly between pH 9 to 10.7 (Iler, 1979) and that is also the reason why digestion of the $bSiO_2$ samples is conducted in an alkaline medium. Additionally, the water-$bSiO_2$ mixture was transferred to a Teflon vial for further handling, so in the unlikely case of fractionation effects during dissolution, the bulk would have been further processed and no isotopic signal was lost.

1) 238. Typo, 2 times « and »

2+3) We removed the second 'and'.

1) 301 Is it worth keeping MUC-22-04 whose bottom SW has been contaminated? (likely by surface SW)

2) Because of completeness of the results section, we prefer to leave the sample MUC-22-04 in and report the Si concentration and $\delta^{30}Si$ values.

1) 366: the kinetics of reverse weathering is poorly known, especially in situ and is certainly not immediate. So do not to use such wording « *as soon as Si is released (...), it reprecipitates* ». Moreover not all Si reprecipitates, otherwise there won't be more DSi in pore fluids than in bottom water (indeed in their previous work, Elhert et al. 2016 have quantified that only 24% of dissolved bSi reprecipitates). This sentence needs to be corrected.

3) We corrected the sentence accordingly (lines 369-372).

1) 371-372 From the three references cited here, only Elhert et al. 2016 has estimated fractionation factors for reverse weathering, the other two refer to continental weathering (Georg et al., 2006 and Opfegerlt et al., 2013). Remove them or specify it since this sentence is misleading.

3) We removed the references *Georg et al., 2009* and *Opfergelt et al., 2013* (lines 374-375).

1) 395-398. Sentence unclear / grammatically incorrect

3) We rewrote the sentence (lines 403-406).

1) 397. At least one reference should be cited for Si isotope fractionation during bSi dissolution (e.g. DeMarest et al., 2009).

2+3) We discussed the potential impact on Si isotope fractionation during dissolution and add the following references: Demarest et al., 2009, Egan et al., 2012, Wetzel et al., 2014 (see also comment to reviewer #1) (lines 397-401).

1) 440-441. This sentence is too affirmative given the level of discussion at this stage of the ms. It could be changed to e.g. "Thus, at Basin sites both K/Al ratios of sediments and the heavier d30Sipf are in agreement to recognise bSi dissolution followed by authigenic clay formation as significant processes taking place".

3) We rephrased the sentence accordingly (lines 450-452).

1) 527: typo in isotope

3) The typo was corrected.

1) Table S5 Typo « auf »

3) The typo was corrected.

1) Fig. 5. It should be mentioned in the caption that Fig. 5b is from another study (Ehlert et al. 2016?)

3) The reference *Ehlert et al., 2016* was included in the caption of Fig. 5.

1) Fig. 6. Need to define red and black dashed line in the caption without having to go through the text in the ms.

3) The definitions of the red and black dashed lines were included in the caption.

**References to the answer letter**

Anderson T. F. and Raiswell R. (2004) SOURCES AND MECHANISMS FOR THE ENRICHMENT OF HIGHLY REACTIVE IRON IN EUXINIC BLACK SEA SEDIMENTS. *Am. J. Sci.* **304**, 203–233.

Van Cappellen P. and Qiu L. Q. (1997) Biogenic silica dissolution in sediments of the Southern Ocean.1. Solubility. *Deep. Res. Part Ii-Topical Stud. Oceanogr.* **44**, 1109–1128.

Ehlert C., Grasse P., Mollier-Vogel E., Böschen T., Franz J., de Souza G. F., Reynolds B. C., Stramma L. and Frank M. (2012) Factors controlling the silicon isotope distribution in waters and surface sediments of the Peruvian coastal upwelling. *Geochim. Cosmochim. Acta* **99**, 128–145.

Frings P. J., Clymans W., Fontorbe G., De La Rocha C. L. and Conley D. J. (2016) The continental Si cycle and its impact on the ocean Si isotope budget. *Chem. Geol.* **425**, 12–36. Available at: http://dx.doi.org/10.1016/j.chemgeo.2016.01.020.

Frings P. J., Rocha C. D. La, Struyf E., Pelt D. Van, Schoelynck J., Hudson M. M., Gondwe M. J., Wolski P., Mosimane K., Gray W., Schaller J. and Conley D. J. (2014) Tracing silicon cycling in the Okavango Delta, a sub-tropical flood-pulse wetland using silicon isotopes. *Geochim. Cosmochim. Acta* **142**, 132–148. Available at: http://www.sciencedirect.com/science/article/pii/S0016703714004694.

Georg R. B., Reynolds B. C., Frank M. and Halliday a. N. (2006) Mechanisms controlling the silicon isotopic compositions of river waters. *Earth Planet. Sci. Lett.* **249**, 290–306.

Georg R. B., Zhu C., Reynolds B. C. and Halliday A. N. (2009) Stable silicon isotopes of groundwater, feldspars, and clay coatings in the Navajo Sandstone aquifer, Black Mesa, Arizona, USA. *Geochim. Cosmochim. Acta* **73**, 2229–2241. Available at: http://dx.doi.org/10.1016/j.gca.2009.02.005.

Iler R. K. (1979) *The Chemistry of Silica*., John Wiley & Sons Inc, New York.

Loucaides S., Michalopoulos P., Presti M., Koning E., Behrends T. and Van Cappellen P. (2010) Seawater-mediated interactions between diatomaceous silica and terrigenous sediments: Results from long-term incubation experiments. *Chem. Geol.* **270**, 68–79. Available at: http://dx.doi.org/10.1016/j.chemgeo.2009.11.006.

Michalopoulos P. and Aller R. C. (2004) Early diagenesis of biogenic silica in the Amazon delta: Alteration, authigenic clay formation, and storage. *Geochim. Cosmochim. Acta* **68**, 1061–1085.

Michalopoulos P., Aller R. C. and Reeder R. J. (2000) Conversion of diatoms to clays during early diagenesis in tropical, continental shell muds. *Geology* **28**, 1095–1098.

Opfergelt S., Georg R. B., Delvaux B., Cabidoche Y. M., Burton K. W. and Halliday a. N. (2012) Silicon isotopes and the tracing of desilication in volcanic soil weathering sequences, Guadeloupe. *Chem. Geol.* **326**–**327**, 113–122. Available at: http://dx.doi.org/10.1016/j.chemgeo.2012.07.032.

Parkhurst B. D. L. and Appelo C. a J. (1999) User's Guide To PHREEQC (version 2) — a Computer Program for Speciation, and Inverse Geochemical Calculations. *Exch. Organ. Behav. Teach. J.* **D**, 326. Available at: http://downloads.openchannelsoftware.org/PHREEQC/manual.pdf.

Reynolds B. C., Frank M. and Halliday A. N. (2008) Evidence for a major change in silicon cycling in the subarctic North Pacific at 2 . 73 Ma. *Paleoceanography* **23(PA4219)**.

Savage P. S., Georg R. B., Williams H. M., Burton K. W. and Halliday A. N. (2011) Silicon isotope fractionation during magmatic differentiation. *Geochim. Cosmochim. Acta* **75**, 6124–6139.

Singer A. (1980) The Paleoclimatic Interpretation of Clay Minerals in Soils and Weathering Profiles. *Earth-Science Rev.* **15**, 303–326.

Wilson M. J. (2004) Weathering of the primary rock-forming minerals: processes, products and rates. *Clay Miner.* **39**, 233–266.

[revised manuscript text omitted]